# Differential chromatin binding of the lung lineage transcription factor NKX2-1 resolves opposing murine alveolar cell fates in vivo

Danielle R. Little[1,2], Anne M. Lynch[1,3], Yun Yan [2], Haruhiko Akiyama[4], Shioko Kimura [5] & Jichao Chen [1✉]

Differential transcription of identical DNA sequences leads to distinct tissue lineages and then multiple cell types within a lineage, an epigenetic process central to progenitor and stem cell biology. The associated genome-wide changes, especially in native tissues, remain insufficiently understood, and are hereby addressed in the mouse lung, where the same lineage transcription factor NKX2-1 promotes the diametrically opposed alveolar type 1 (AT1) and AT2 cell fates. Here, we report that the cell-type-specific function of NKX2-1 is attributed to its differential chromatin binding that is acquired or retained during development in coordination with partner transcriptional factors. Loss of YAP/TAZ redirects NKX2-1 from its AT1-specific to AT2-specific binding sites, leading to transcriptionally exaggerated AT2 cells when deleted in progenitors or AT1-to-AT2 conversion when deleted after fate commitment. *Nkx2-1* mutant AT1 and AT2 cells gain distinct chromatin accessible sites, including those specific to the opposite fate while adopting a gastrointestinal fate, suggesting an epigenetic plasticity unexpected from transcriptional changes. Our genomic analysis of single or purified cells, coupled with precision genetics, provides an epigenetic basis for alveolar cell fate and potential, and introduces an experimental benchmark for deciphering the in vivo function of lineage transcription factors.

[1] Department of Pulmonary Medicine, The University of Texas MD Anderson Cancer Center, Houston, TX, USA. [2] The University of Texas MD Anderson Cancer Center UTHealth Graduate School of Biomedical Sciences, Houston, TX, USA. [3] Graduate Program in Developmental Biology, Baylor College of Medicine, Houston, TX, USA. [4] Department of Orthopaedics, Gifu University, Gifu City, Japan. [5] Laboratory of Metabolism, Center for Cancer Research, National Cancer Institute, National Institutes of Health, Bethesda, MD, USA. ✉email: jchen16@mdanderson.org

Organism development requires sequential choreographed restriction of progenitors to distinct tissue lineages, marked by lineage transcription factors, and subsequently cell types within each lineage. This progressive restriction is exemplified by the formation of lung, pancreas, and hindgut epithelia from the embryonic endoderm—which are marked and controlled by NKX2-1, PDX1, and CDX2, respectively—and further differentiation into distinct secretory and absorptive cell types[1–3]. Unlike lineages of a single-cell type such as skeletal muscle cells, marked by MYOD[4], and melanocytes, marked by MITF[5], little is known about whether and how lineage transcription factors function distinctly in diverse and often opposing cell types within the same lineage. This is because, unlike cell culture or temporal shift in cell fate, coexisting cell types in native tissues must be purified to obtain interpretable epigenomic data[6–8]. Deciphering such cell-type-specific functions of lineage transcription factors will shed light on their integration with spatiotemporal inputs of less unique transcription factors to determine cell fates during development and regeneration[7,8].

The mouse lung alveolar epithelium provides a robust system to study its lineage transcription factor NKX2-1, as its constituent cell types can be precisely targeted genetically and purified in millions for genomic analysis. Specifically, alveolar type 1 (AT1) and alveolar type 2 (AT2) cells are polar opposites in function and morphology with underlying distinct gene expression. AT1 cells are extremely thin yet expansive, allowing passive gas diffusion, whereas AT2 cells are cuboidal and secret surfactants to reduce surface tension[1]. Nevertheless, both cell types arise from embryonic SOX9 progenitors, and thus their opposing cell fates must be resolved during development and their identities guarded afterwards, especially given that AT2 cells are able to differentiate into AT1 cells during injury-repair[9,10]. Both AT1 and AT2 cells express NKX2-1 and whole lung ChIP-seq shows that NKX2-1 binds to both AT1 and AT2-specific genes[11]. However, cell-type-specific deletion experiments demonstrate that NKX2-1 is cell-autonomously required for AT1 and AT2 cell differentiation, raising the question of how the same lineage transcription factor regulates cell-type-specific genes in each cell type.

Conceptually, NKX2-1 and transcription factors in general recognize short DNA sequences, known as motifs, which occur too frequently in the mammalian genome to be informative on their own, thereby also limiting the utility of in vitro and even in vivo reporter assays as well as overexpression models[6]. Additional specificity arises from neighboring sequences that are bound by partner transcription factors and from differential chromatin accessibility regulated by pioneer factors and chromatin remodelers[12,13]. Furthermore, transcription factor binding does not necessarily result in transcriptional changes in nearby genes, reflecting instead primed/poised enhancers, shadow enhancers, or simply opportunistic binding[6,14,15]. The relevance of these modes of action to a given transcription factor ideally needs to be defined in native tissues using purified cell populations, since DNA-binding and epigenetic features differ widely across cell types.

In this study, using both bulk and single-cell transcriptomic and epigenomic analyses of purified native cells, we map NKX2-1 binding in AT1 versus AT2 cells, as well as their embryonic progenitors; examine the effect of NKX2-1 binding on chromatin accessibility using cell-type-specific Nkx2-1 mutants; and identify and functionally test partner transcriptional factors. We show that in native tissues, the lineage transcription factor NKX2-1 resolves opposite cell fates and exerts cell-type-specific functions via differential binding in part under the control of YAP/TAZ transcriptional cofactors, and that NKX2-1 binding regulates a cell-type-specific epigenetic landscape that is not predicted by the transcriptome, providing insights into cell fate determination during lung development and injury-repair.

## Results

**NKX2-1 binds chromatin in a cell-type-specific manner.** The cell-type-specific function of a transcription factor—in this case, NKX2-1 in AT1 versus AT2 cell differentiation—can be attributed to differential chromatin binding or equal binding but differential transcriptional activity. NKX2-1 ChIP-seq experiments using whole lungs have demonstrated the ability of NKX2-1 to bind to both AT1 and AT2 genes, but cannot distinguish the two aforementioned mechanisms[11]. Therefore, we adopted a cell-type-specific epigenomic method[16] and performed ChIP-seq for NKX2-1 and various histone marks using purified GFP-expressing nuclei that were genetically labeled by either a newly characterized AT1-specific driver Wnt3a[Cre17] or an AT2-specific driver Sftpc[CreER18]. We showed that the Wnt3a[Cre] driver exhibited 76% efficiency and 100% specificity within the epithelium of 10-week-old lungs (4215 GFP+ cells from 3 mice) and 62% efficiency and 99.3% specificity at postnatal day (P) 7 (2096 GFP+ cells from 2 mice) (see Source Data for all raw cell counts). The non-AT1 cells targeted by Wnt3a[Cre] were immune cells that accumulated over time to range from 3.3% of GFP+ cells at P7 (Supplementary Fig. 1c) to 30% at 10-week-old, but did not express NKX2-1. We confirmed that the Sftpc[CreER] driver exhibited 92% efficiency and 99.8% specificity within the epithelium of 10-week-old lungs (1698 GFP+ cells from 3 mice) and 97% efficiency and 96% specificity at P7 (2096 GFP+ cells from 3 mice) (Fig. 1a and Supplementary Fig. 1). Comparison of the H3K4me3 signal, a marker of active promoters[19], in purified AT1 or AT2 nuclei with data from the whole lung validated the expected enrichment and depletion of H3K4me3 near an AT1 (Spock2) or AT2 (Lamp3) gene, a generic epithelial gene (Cdh1; also known as E-Cadherin), and a mesenchymal gene (Pdgfra) (Fig. 1b) (see Source Data for all representative cell sorting schemes).

NKX2-1 ChIP-seq using purified AT1 and AT2 nuclei from adult lungs revealed three categories of NKX2-1 sites: AT1-specific, AT2-specific, and common to both (Fig. 1c). Cell-type-specific NKX2-1 sites were often near corresponding cell-type-specific genes and associated with enrichment of H3K4me3 (active promoter) and H3K27ac and H3K4me1 (putative enhancer), as exemplified by an AT1-specific (Spock2) or AT2-specific (Lamp3) gene (Fig. 1c, d and Supplementary Data 1). The majority of cell-type-specific NKX2-1 sites were distal regulatory elements (>2 kb from the nearest defined transcription start site; 92% and 89% for AT1-specific and AT2-specific sites, respectively), as also reflected in the enrichment of H3K27ac signals relative to H3K4me3 signals (Fig. 1c).

We posited that the common NKX2-1 sites were associated with characteristics shared between AT1 and AT2 cells and thus could include lung epithelial lineage sites or those accessible in nearly all cell types, which we termed housekeeping sites by analogy to housekeeping genes. To define and deconvolve these lineage and housekeeping sites, we performed single-cell ATAC-seq (scATAC-seq) on adult lung cells of the four lineages—epithelial, endothelial, immune, and mesenchymal—that were purified and combined in equal proportions to adequately sample multiple cell types within each lineage as described[20] (Supplementary Fig. 2a). Of the NKX2-1 binding sites common between AT1 and AT2 cells, 5862 peaks had higher accessibility in the epithelial lineage and were considered lineage sites, as exemplified by Cdh1, whereas the remaining 20,881 peaks were accessible in all four lineages and were considered housekeeping sites, as exemplified by Gapdh (Fig. 1c, d, Supplementary Fig. 2b, c, and Supplementary Data 1). A subset of the lineage sites (2221 peaks)

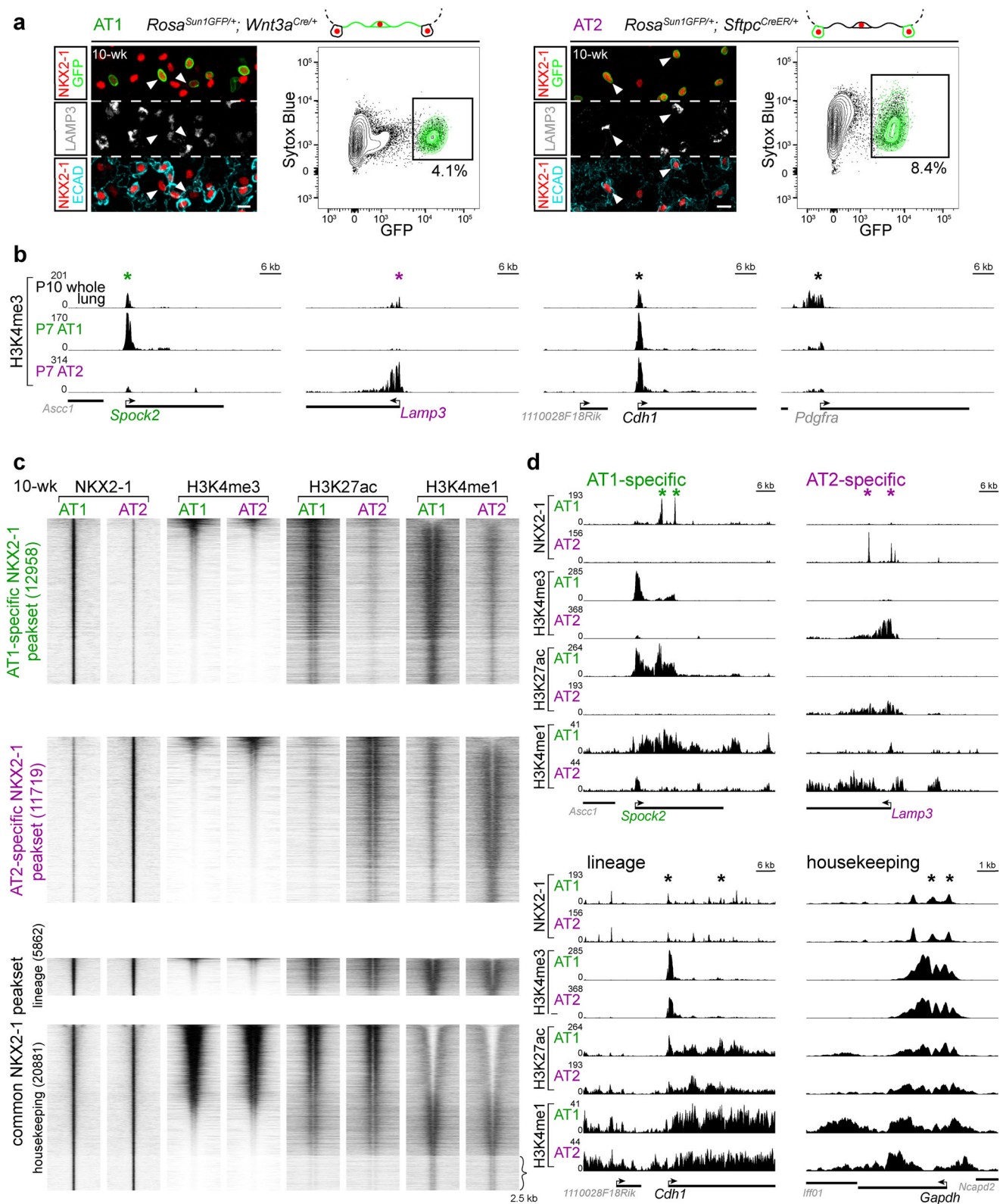

had lower accessibility in SOX2-expressing airway cells, likely reflecting the distinct anatomic location and developmental history of alveolar versus airway cells[21] (Supplementary Fig. 2b and Supplementary Data 1).

Like the cell-type-specific sites, NKX2-1 lineage sites were mostly distal to promoters (91.9%), whereas the housekeeping sites were often found near promoters (42%), as supported by the histone marks (Fig. 1c). Notably, 21% of sites in the housekeeping category were depleted for all three histones marks (Fig. 1c) and were associated with a 36-fold (60% over 1.7% background) enrichment for the CTCF motif, indicative of chromatin insulators[22], in addition to the expected NKX motif (62% over

**Fig. 1 NKX2-1 binds chromatin in a cell-type-specific manner. a** Genetic labeling and FACS purification of AT1 and AT2 nuclei using *Wnt3a*^Cre and *Sftpc*^CreER, respectively. Confocal images of immunostained lungs showing that both AT1 and AT2 cells express NKX2-1 while AT2 cells are distinguishable by LAMP3 and cuboidal E-Cadherin (ECAD). *Rosa*^Sun1GFP marks the nuclear envelop (arrowhead; green in the diagram). All nuclei are Sytox Blue positive. 10-wk, 10-week-old. For *Sftpc*^CreER, 3 mg tamoxifen was administrated 3 days before tissue harvest. Scale: 10 μm. The color scheme (AT1: green; AT2: purple) applies to subsequent figures. **b** Validation of cell-type-specific ChIP-seq using an active promoter marker H3K4me3. Compared to the whole lung, the site (asterisk; same in subsequent figures) near an AT1 gene *Spock2* shows enrichment in purified AT1 nuclei but depletion in purified AT2 nuclei, whereas the site near an AT2 gene *Lamp3* shows the opposite pattern. A pan-epithelial gene *Cdh1* (also known as *E-Cadherin*) is enriched in both AT1 and AT2 nuclei, whereas a mesenchymal gene *Pdgfra* is depleted in both. P, postnatal day. For *Sftpc*^CreER, 250 μg tamoxifen was administrated 3 days before tissue harvest. **c** ChIP-seq heatmaps (2.5 kb flanking the peak center; same in subsequent figures) of purified AT1 and AT2 nuclei for NKX2-1, H3K4me3 (active promoter), H3K27ac (putative active enhancer; note its bimodal pattern surrounding the center), and H3K4me1 (putative enhancer and active gene body), grouped by differential NKX2-1 binding into AT1-specific, AT2-specific, and common peaksets and sorted by the H3K4me3 signal. The common peakset is divided into lineage (epithelial) and housekeeping sets, based on scATAC-seq (Supplementary Fig. 2). Compared to the cell-type-specific and lineage sets, the housekeeping set has abundant H3K4me3, which also corresponds to depletion of H3K4me1 (top portion), whereas the bottom portion (curly bracket) is depleted of H3K4me3 and is enriched for the insulator CTCF motif. **d** Example NKX2-1 binding sites of the AT1-specific (*Spock2*), AT2-specific (*Lamp3*), lineage (*Cdh1*), and housekeeping (*Gapdh*) sets. The Y-axes are scaled via foreground normalization so that peak heights can be directly compared across samples; quantification and statistics are reported in the corresponding Supplementary Data files with marked peaks highlighted (same in subsequent figures).

---

34% background), implicating NKX2-1 in higher order chromatin organization. Together, these data delineate the common and distinct binding profiles of NKX2-1 in AT1 versus AT2 cells and suggest that NKX2-1 regulates the opposing cell fates via cell-type-specific binding.

**NKX2-1 is required for accessibility at cell-type-specific sites.** To test the functionality of the identified NKX2-1 binding, we examined changes in chromatin accessibility 5 days after AT1 or AT2-specific *Nkx2-1* deletion in the adult lung using a newly generated driver under the control of an AT1-specific gene *Rtkn2*^CreER (95% efficiency and 98% specificity based on 1356 GFP+ cells from 3 mice; Supplementary Fig. 3a, b, e) or *Sftpc*^CreER (95% efficiency and 99.8% specificity based on 4215 GFP+ cells from 3 mice), respectively. In the resulting *Nkx2-1*^CKO/CKO; *Rosa*^Sun1GFP/+; *Rtkn2*^CreER/+ mutant (abbreviated as NKX2-1^Rtkn2), NKX2-1 was lost in 86% of recombined GFP-expressing cells at 5 days after recombination (1339 GFP+ cells from 3 mice) and 99% at 7 days after recombination (based on 1011 GFP+ cells from 3 mice) (Fig. 2a). An AT1-specific marker HOPX was lost, and a proliferation marker KI67 was ectopically expressed in occasional cells 5 days after recombination but in 51% of GFP+ cells 7 days after recombination (Fig. 2a and Supplementary Fig. 3c). Similarly, in the *Nkx2-1*^CKO/CKO; *Rosa*^Sun1GFP/+; *Sftpc*^CreER/+ mutant (abbreviated as NKX2-1^Sftpc), NKX2-1 (82% of 6570 GFP+ cells from 3 mice) and SFTPC were lost, while KI67 was ectopically expressed 5 days after recombination (34% of 6570 GFP+ cells from 3 mice) (Fig. 2b and Supplementary Fig. 3d).

Bulk ATAC-seq comparison of purified GFP-expressing AT1 cells from the NKX2-1^Rtkn2 mutant and littermate control lungs showed an obvious reduction in accessibility at AT1-specific NKX2-1 binding sites and lineage sites, but limited changes at AT2-specific and housekeeping sites (Fig. 2c, d and Supplementary Data 1). Conversely, purified AT2 cells from the NKX2-1^Sftpc mutant lost accessibility at AT2-specific NKX2-1 binding sites and lineage sites, but not AT1-specific and housekeeping sites (Fig. 2c, d and Supplementary Data 1). We noted that mutant AT1 cells had a larger decrease in accessibility than mutant AT2 cells at lineage sites such as *Cdh1*—perhaps reflecting differential sensitivity of some genes to loss of NKX2-1 (Fig. 2d). *Sftpb*, an AT2 gene, had both AT2-specific and lineage NKX2-1 sites that depended on NKX2-1 in the expected manner (Fig. 2d), suggesting combinatorial control by multiple regulatory elements. Notwithstanding gene-specific differences, accessibility at cell-type specific NKX2-1 peaksets in both AT1 and AT2 cells

depends upon NKX2-1, supporting the idea that such binding is functional at least with regard to chromatin accessibility.

**NKX2-1 establishes cell-type-specific binding via selectively acquiring de novo sites and retaining sites bound in progenitors.** Next, we sought to determine the kinetics of establishing AT1 and AT2-specific NKX2-1 binding sites as AT1 and AT2 cells differentiate from their SOX9-expressing progenitors[21]. We chose to profile SOX9 progenitors at embryonic day (E) 14.5 when no appreciable AT1 versus AT2 differentiation was detectible by single-cell RNA-Seq (scRNA-seq)[23]. Although we used E14.5 whole lungs, instead of purified cells, to obtain enough tissue, NKX2-1 was not present in non-epithelial cells and SOX9 progenitors constituted the majority (70%) of epithelial cells[23]. Since AT1 cells continued to grow after birth[24], we additionally profiled purified AT1 and AT2 cells at P7 to capture their intermediate epigenetic states using *Wnt3a*^Cre and *Sftpc*^CreER, respectively.

We found that NKX2-1 binding sites that were differential in the adult AT1 versus AT2 cells could range from absent to abundant in the progenitors: the 20% least present sites showed a gradual increase in NKX2-1 binding over time in the expected cell type with little increase in the alternative cell type, which we termed acquired sites; whereas the 20% most present sites maintained NKX2-1 binding in the expected cell type but showed a gradual loss in the alternative cell type, which we termed retained sites (Fig. 3a, b and Supplementary Data 2). Therefore, as additionally exemplified by *Pdpn* and *Hopx* for AT1-specific sites and *Sftpb* and *Lamp3* for AT2-specific sites (Fig. 3c, d), cell-type-specific NKX2-1 binding is achieved via selectively acquiring sites de novo as well as by retaining sites already bound in progenitors. Quantification of such kinetics confirmed the gradual increase or decrease in binding over time (Fig. 3a, e) and also suggested that the binary on/off binding in individual cells is graduated on a population level, likely due to asynchrony across cells/genes and/or varied duration in binding within a cell, providing a possible molecular explanation for gradual alveolar cell maturation.

Unlike cell-type-specific NKX2-1 sites, of which a comparable number were acquired and retained, sites common to AT1 and AT2 cells were mostly retained, as exemplified by *Irf2*, consistent with them being lineage and housekeeping sites (Supplementary Fig. 4a–c and Supplementary Data 2). Our kinetics analysis also identified 6933 progenitor-specific NKX2-1 binding sites that decreased quickly or slowly, but similarly along the paths of AT1 versus AT2 differentiation, as exemplified by *Tinag* (Supplementary Fig. 4a–c).

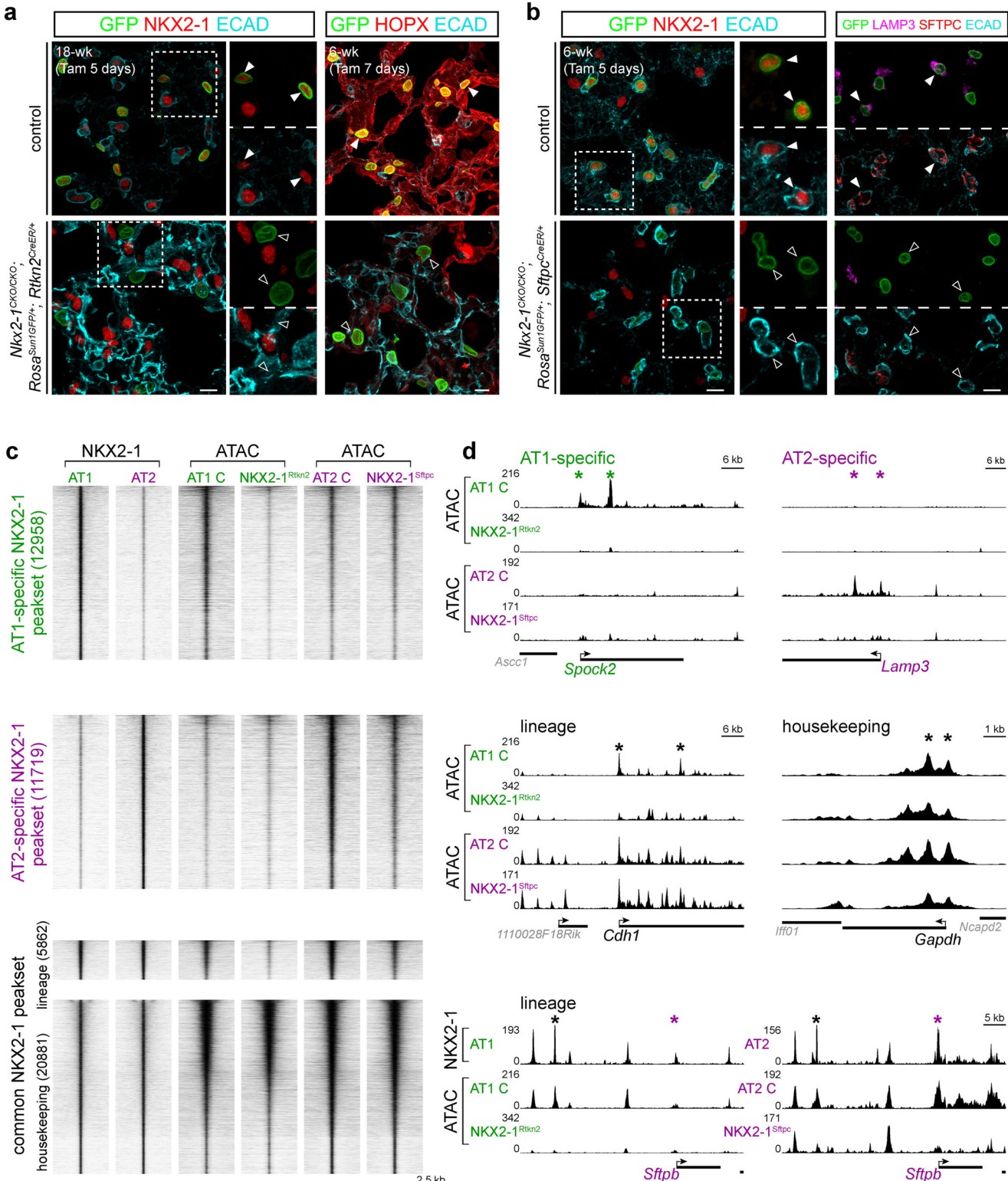

**Acquired and retained NKX2-1 sites have comparable kinetics in transcriptional divergence of progenitors toward opposing cell fates.** The two modes of achieving differential binding— acquiring de novo sites versus retaining bound sites—prompted us to examine if these modes were associated with distinct kinetics of transcription. We compiled 12 scRNA-seq datasets from E14.5 to 15-week-old lungs that included 23,577 epithelial cells (Fig. 4a and Supplementary Fig. 4d). AT1 and AT2 cells were readily

distinguishable at E18.5 but clustered separately from their post-natal counterparts, perhaps due to transcriptional changes upon exposure to air and airborne factors (Fig. 4a). Monocle analysis showed a bifurcated trajectory originating from embryonic SOX9 progenitors and splitting toward adult AT1 and AT2 cells, as respectively exemplified by *Sox9*, *Spock2*, and *Lamp3* (Fig. 4b, c). Although SOX9 progenitors continuously exit branch tips and become AT1 and AT2 cell precursors at E16.5[25,26], such early

**Fig. 2 NKX2-1 is required for accessibility at cell-type-specific sites.** AT1-specific (**a**) and AT2-specific (**b**) deletion of *Nkx2-1* using *Rtkn2*^CreER^ and *Sftpc*^CreER^, respectively. Confocal images of immunostained mature lungs, showing efficient loss (filled versus open arrowhead) of NKX2-1 in recombined cells marked by GFP (**a** and **b**), as well as an AT1 marker (HOPX in **a**) and AT2 markers (LAMP3 and SFTPC in **b**). Note the increased ECAD staining in the AT1-specific *Nkx2-1* mutant (**a**), consistent with our publication[11]. Two doses (3 mg each; 48 h interval) of tamoxifen (Tam) were given and the lungs were harvested at 5 or 7 days after the initial dose. Scale: 10 μm. **c** ATAC-seq heatmaps of purified AT1 cells from AT1-specific *Nkx2-1* mutant (as in **a**; abbreviated as NKX2-1^Rtkn2^) and littermate control (AT1 C) lungs, and purified AT2 cells from AT2-specific *Nkx2-1* mutant (as in **b**; abbreviated as NKX2-1^Sftpc^) and littermate control (AT2 C) lungs, at the same NKX2-1 ChIP-seq peaksets organized as in Fig. 1. Both mutants are from mature lungs at 5 days after the initial tamoxifen administration as in **a**. AT1-specific and AT2-specific NKX2-1 peaksets have reduced accessibility in the corresponding cell type and cell-type-specific mutants; the lineage peakset has reduced accessibility in both mutants; whereas the housekeeping peakset is largely unaffected. **d** Example chromatin accessibility signals at NKX2-1 binding sites of the AT1-specific (*Spock2*), AT2-specific (*Lamp3*), lineage (*Cdh1*), and housekeeping (*Gapdh*) sets. Accessibility at sites near *Cdh1* in AT2 cells is unaffected by *Nkx2-1* deletion, possibly due to AT2-specific redundant mechanisms to maintain accessibility. An NKX2-1-bound lineage site ~30 kb upstream of *Sftpb* (black asterisk) has reduced accessibility in both *Nkx2-1* mutants. *Sftpb* also has an AT2-specific NKX2-1 site at its promoter (purple asterisk) that has reduced accessibility specifically in the NKX2-1^Sftpc^ mutant.

molecular differentiation was placed by Monocle within the progenitor branch and prior to the AT1/AT2 divergence (Fig. 4b, c).

To buffer the known uncertainty in attributing regulatory elements to target genes[27], we assigned all acquired or retained cell-type-specific sites to their nearest gene and generated an expression score averaging over all genes within a set (>1000 genes)—an averaging approach deployed in cell-cycle scoring in Seurat or gene set enrichment analysis[28,29]. The resulting module score for the acquired sites was low in SOX9 progenitors and increased along the trajectory toward the expected cell type, but remained low in the alternative cell type (Fig. 4d, e and Supplementary Data 3). Surprisingly, the module score for the retained sites had essentially the same kinetics, despite a high level of NKX2-1 binding in SOX9 progenitors (Fig. 4d, e and Supplementary Data 3). By comparison, acquired or retained sites that were common to AT1 and AT2 cells had a score that increased or did not change, respectively, along both AT1 and AT2 cell trajectories (Supplementary Fig. 4e, f and Supplementary Data 3). Module scores for progenitor-specific NKX2-1 sites did not change, perhaps due to exclusion of proliferating cells from our Monocle analysis (Supplementary Fig. 4e, f and Supplementary Data 3). The remarkable dissociation between NKX2-1 binding and gene expression for the retained cell-type-specific sites (Fig. 4d, e) suggested that the epigenome of the progenitors is earmarked by NKX2-1 for future transcription of cell-type-specific genes, providing epigenetic evidence for SOX9 progenitors being bipotential progenitors for AT1 and AT2 cells[10,21].

**AT1-specific partner factors YAP/TAZ establish AT1-specific NKX2-1 binding and cell fate, and antagonize those of AT2 cells.** The functional, differential NKX2-1 binding in AT1 versus AT2 cells led us to pursue the hypothesis that cell-type-specific NKX2-1 binding was guided by partner transcription factors, which should be cell-type-specific and have their binding sites near those of NKX2-1. Accordingly, we performed HOMER de novo motif analysis of AT1-specific, AT2-specific, and common NKX2-1 sites (Fig. 5a) and found the expected NKX motif in all three categories (54% over 25% background, 58% over 22% background, and 52% over 27% background, respectively). Intriguingly, the second most enriched motif was TEAD for AT1-specific sites and CEBP for AT2-specific sites, whereas the common sites contained motifs for FOXA, likely corresponding to the endoderm regulators FOXA1/A2[30], as well as CTCF, an insulator factor as described earlier (Fig. 5a). To pinpoint the specific members of the TEAD motif family, we used our scRNA-seq dataset and found that *Tead1/4* were enriched in AT1 cells (Supplementary Fig. 5a). Anticipating the complex genetics required to dissect possible redundancy among the four TEAD homologs, we focused on their obligatory Hippo signaling cofactors, YAP/TAZ, because the canonical target genes, *Ctgf* and

*Cyr61*[31], were specific to AT1 cells (Supplementary Fig. 5a). Indeed, active nuclear YAP/TAZ were specifically detected in developing and mature AT1 cells (Fig. 5b). Similarly, CEBPA was specific to AT2 cells on both transcriptional and protein levels (Fig. 5b and Supplementary Fig. 5a).

To test if YAP/TAZ/TEAD functioned as partner transcription factors for NKX2-1, we deleted *Yap/Taz* from SOX9 progenitors using our previously characterized *Sox9*^CreER[11,32] at E15.5 when AT1 and AT2 cell differentiation just started, and performed NKX2-1 ChIP-seq on E18.5 control and *Yap/Taz* mutant (abbreviated as Y/T^Sox9^) whole lungs (Fig. 5c, d and Supplementary Data 4). As we hypothesized, 5877 sites with decreased NKX2-1 binding corresponded to AT1-specific NKX2-1 sites in the adult lung, as exemplified by sites near *Spock2* and *Hopx* (Fig. 5d, e). Interestingly, 5276 sites had an increase in NKX2-1 binding in the Y/T^Sox9^ mutant and corresponded to AT2-specific NKX2-1 sites, as exemplified by sites near *Lamp3* and *Cebpa* (Fig. 5d, e). Furthermore, 73% of the sites with decreased NKX2-1 binding due to loss of YAP/TAZ had TEAD motifs and the average distance between TEAD and NKX motifs were 52 base pairs, as opposed to 35% and 79 base pairs for unaffected sites and 23% and 97 base pairs for sites with increased NKX2-1 binding (Supplementary Data 4). These biases in the co-occurrence and spacing between TEAD and NKX motifs were consistent with the possibility that YAP/TAZ/TEAD and NKX2-1 exist in a transcription regulatory complex, as suggested by cell culture and human genetics studies[33,34]. Collectively, YAP/TAZ and by extension TEADs direct NKX2-1 to its AT1-specific sites and prevent its binding to AT2-specific sites, at least on the population level.

To determine the transcriptomic consequence of such a shift in NKX2-1 binding and to provide resolution on the level of individual cells, we performed scRNA-seq on E18.5 Y/T^Sox9^ mutant and littermate control lungs. The Y/T^Sox9^ mutant lung had many fewer AT1 cells accompanied by a large increase in AT2 cells (Fig. 5f); the remaining AT1 cells upregulated a subset of AT2 genes, such as *Sftpc* and *Lamp3*, compared to their counterparts in the control lung (Fig. 5g and Supplementary Data 5). Control AT2 cells at E18.5 expressed a low level of AT1 genes, such as *Hopx* and *Ager*, likely due to perdurance from SOX9 progenitors during their initial differentiation toward AT1 cells and/or incomplete silencing of AT1 genes as a feature of future stem cells. Such remnant expression was further reduced in mutant AT2 cells, whereas a subset of AT2 genes, such as *Il33* and *Lcn2*, were increased, suggesting the formation of transcriptionally "exaggerated" AT2 cells in the absence of YAP/TAZ (Fig. 5g and Supplementary Fig. 5b, c). Such exaggerated differentiation was evident in a Monocle trajectory analysis to capture the associated transcriptomic shift, showing ectopic appearance of cells and associated genes, such as *Il33*, beyond the

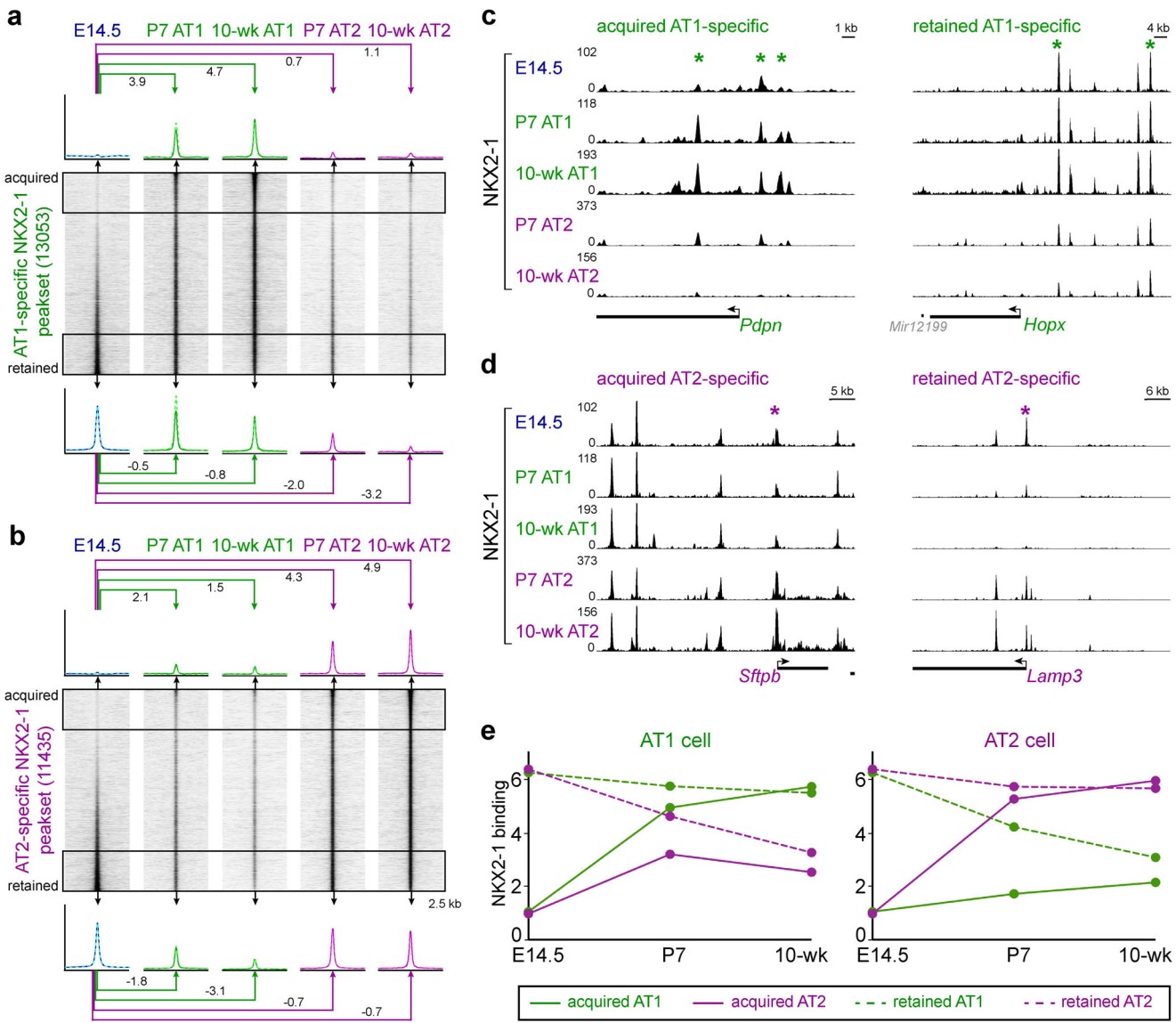

**Fig. 3 NKX2-1 establishes cell-type-specific binding via selectively acquiring de novo sites and retaining sites bound in progenitors.** NKX2-1 ChIP-seq heatmaps from E14.5 whole lungs, purified AT1 or AT2 nuclei from P7 or 10-week-old lungs, grouped by differential NKX2-1 binding in 10-week-old AT1 versus AT2 cells into AT1-specific (**a**) and AT2-specific (**b**) peaksets and sorted by differential NKX2-1 binding between E14.5 lungs and 10-week-old AT1 (**a**) or AT2 (**b**) cells. The top (acquired) and bottom (retained) 20% sites (boxed area) are averaged for the profile plots (dash represents the biological replicate) and the corresponding log2 fold changes from the E14.5 lungs. Acquired AT1-specific sites have little binding in progenitors, and increase over time in AT1 but not AT2 cells; whereas retained AT1-specific sites have high binding in progenitors, and remain high in AT1 but not AT2 cells (**a**). The same kinetics applies to AT2-specific sites (**b**). The small difference in the number of peaks from Fig. 1 is due to incorporation of E14.5 and P7 NKX2-1 ChIP-seq data. Example NKX2-1 binding sites of acquired (*Pdpn*) or retained (*Hopx*) AT1-specific sets (**c**) and acquired (*Sftpb*) or retained (*Lamp3*) AT2-specific sets (**d**). **e** NKX2-1 binding signal of the indicated four categories in **a** and **b**, showing the kinetics leading to differential binding at cell-type-specific sites as AT1 (left) versus AT2 (right) cells mature.

normal AT2 cells in the Y/T$^{Sox9}$ mutant (Fig. 5h, i and Supplementary Data 5). This linear trajectory was different from the bifurcated one that included SOX9 progenitors (Fig. 4b), suggesting that Y/T$^{Sox9}$ mutant cells were not arrested as progenitors, but differentiated toward and even past the embryonic AT2 cell fate. This was substantiated by a module score analysis of genes associated with the exaggerated AT2 cells, showing a high score specifically in AT2 cells among all other major lung cell types (Supplementary Fig. 5d). The loss of AT1 cell fate was confirmed by immunostaining for HOPX and PDPN, consistent with prior *Yap/Taz* mutant phenotypes[35], while most cells in the mutant lung expressed AT2 markers including SFTPC and LAMP3 (Supplementary Fig. 5e).

To relate the transcriptomic shift upon *Yap/Taz* deletion to the change in NKX2-1 binding, we assigned the 20% most decreased or increased NKX2-1 sites to their nearest gene and derived an average expression score for each gene set and plotted them along the Monocle trajectory, as in our temporal analysis (Fig. 4). The resulting module score for the decreased sites trended lower toward the exaggerated AT2 cells, suggesting that loss of NKX2-1 binding correlated with and likely contributed to gene down-regulation (Fig. 5j and Supplementary Data 5). Conversely, increased NKX2-1 binding upon *Yap/Taz* deletion likely underlay exaggerated AT2 differentiation (Fig. 5j and Supplementary Data 5). Taken together, as predicted by our motif analysis, YAP/TAZ/TEAD indeed establishes AT1-specific NKX2-1 binding,

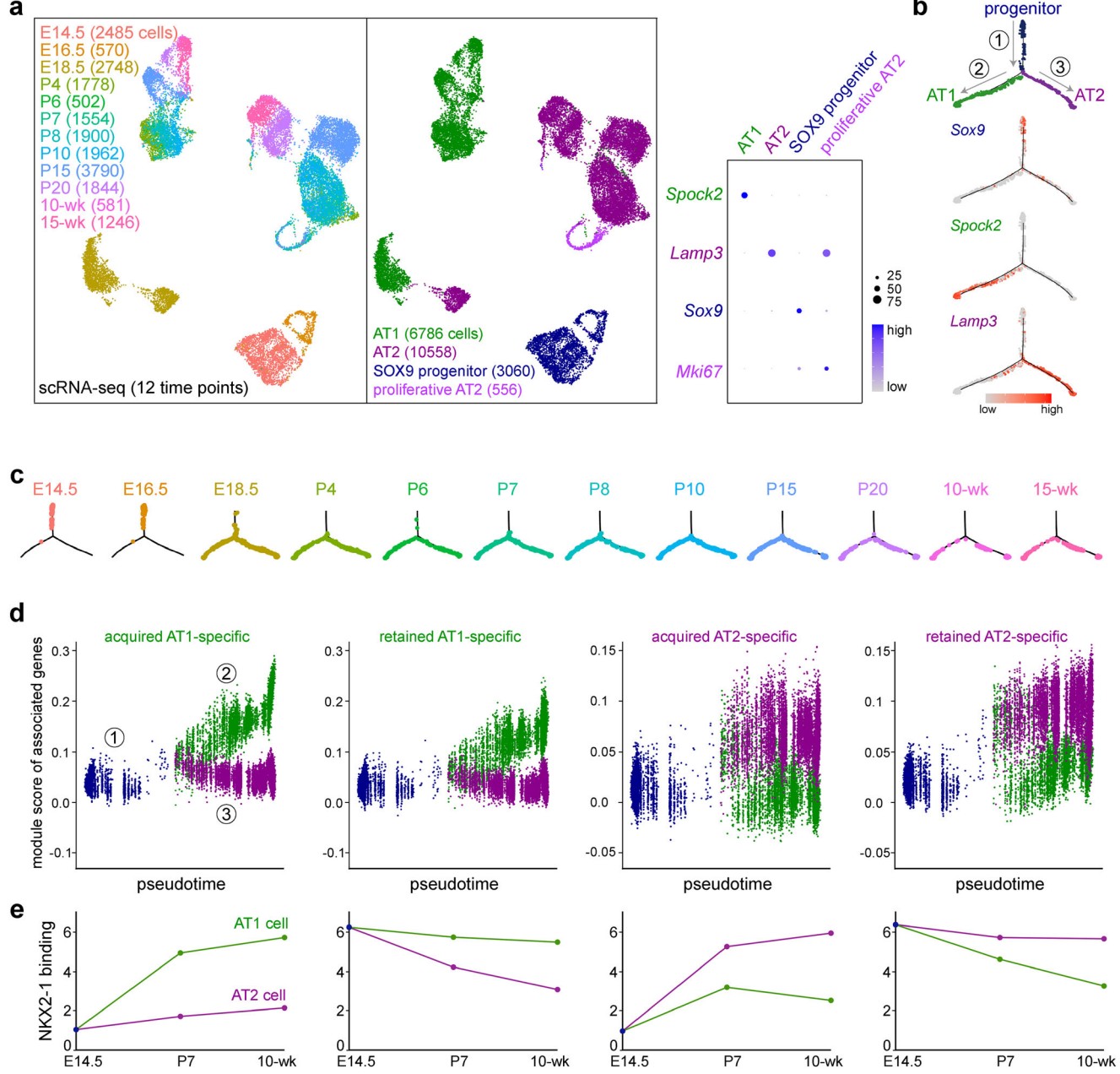

**Fig. 4 Acquired and retained NKX2-1 sites have comparable kinetics in transcriptional divergence of progenitors toward opposing cell fates. a** ScRNA-seq UMAPs (Uniform Manifold Approximation and Projection) of color-coded E14.5 to 15-week-old lungs identify the indicated cell types and numbers in the alveolar region and show the gradual transcriptomic shifts of AT1 and AT2 cells from SOX9 progenitors, as supported by the dot plot for AT1 (*Spock2*), AT2 (*Lamp3*), progenitor (*Sox9*), and proliferative (*Mki67*) cells. Cells from a similar developmental stage (P4, P6, P7, and P8) are clustered together, suggesting the differences among shifted clusters are largely biological. **b** Monocle trajectory analysis of cells in **a** excluding proliferative AT2 cells, showing three branches (circled number) consistent with SOX9 progenitors differentiating into AT1 or AT2 cells, as supported by expression of the respective markers *Sox9*, *Spock2*, and *Lamp3*. The same Monocle trajectory in **c** split by sample time points and colored as in **a**. E14.5 and E16.5 cells are largely homogeneous and limited to branch 1 (progenitor branch). **d** Seurat module scores of gene sets associated with acquired or retained AT1 or AT2-specific NKX2-1 sites as defined in Fig. 3, plotted along the Monocle trajectories as numbered in **c**. **e** NKX2-1 binding signal of the four categories as in **d** in the AT1 or AT2 cell over time, using data from Fig. 3, showing that acquired and retained sites have distinct binding kinetics but comparable transcriptional kinetics.

allowing progenitors to progress toward AT1 cells. Formation of exaggerated AT2 cells in the Y/T$^{Sox9}$ mutant suggested that resolving AT1 versus AT2 cell fate is a gradual process by resisting differentiation toward the opposing cell fate so that, without YAP/TAZ—the "pro-AT1" factors, progenitors accelerate toward the AT2 fate. Strikingly, although tamoxifen interfered with pregnancy and pups were born one day overdue, all Y/T$^{Sox9}$ mutant pups were cyanotic and dead except for one that was gasping—a sign of respiratory distress (Source Data).

**YAP/TAZ maintain AT1-specific NKX2-1 binding and cell fate, and prevent AT1-to-AT2 conversion.** In the Y/T$^{Sox9}$ mutant model, both AT1 and AT2 cells were targeted as descendants of SOX9 progenitors, and NKX2-1 ChIP-seq was performed using whole lungs. To pinpoint the role of YAP/TAZ specifically in AT1 cells and to test whether YAP/TAZ continued to function as partner factors for NKX2-1 after cell fate specification, we generated *Yap/Taz*$^{CKO/CKO}$; *Rosa*$^{Sun1GFP/+}$; *Wnt3a*$^{Cre/+}$ mutants (abbreviated as Y/T$^{Wnt3a}$) and performed NKX2-1 ChIP-seq using

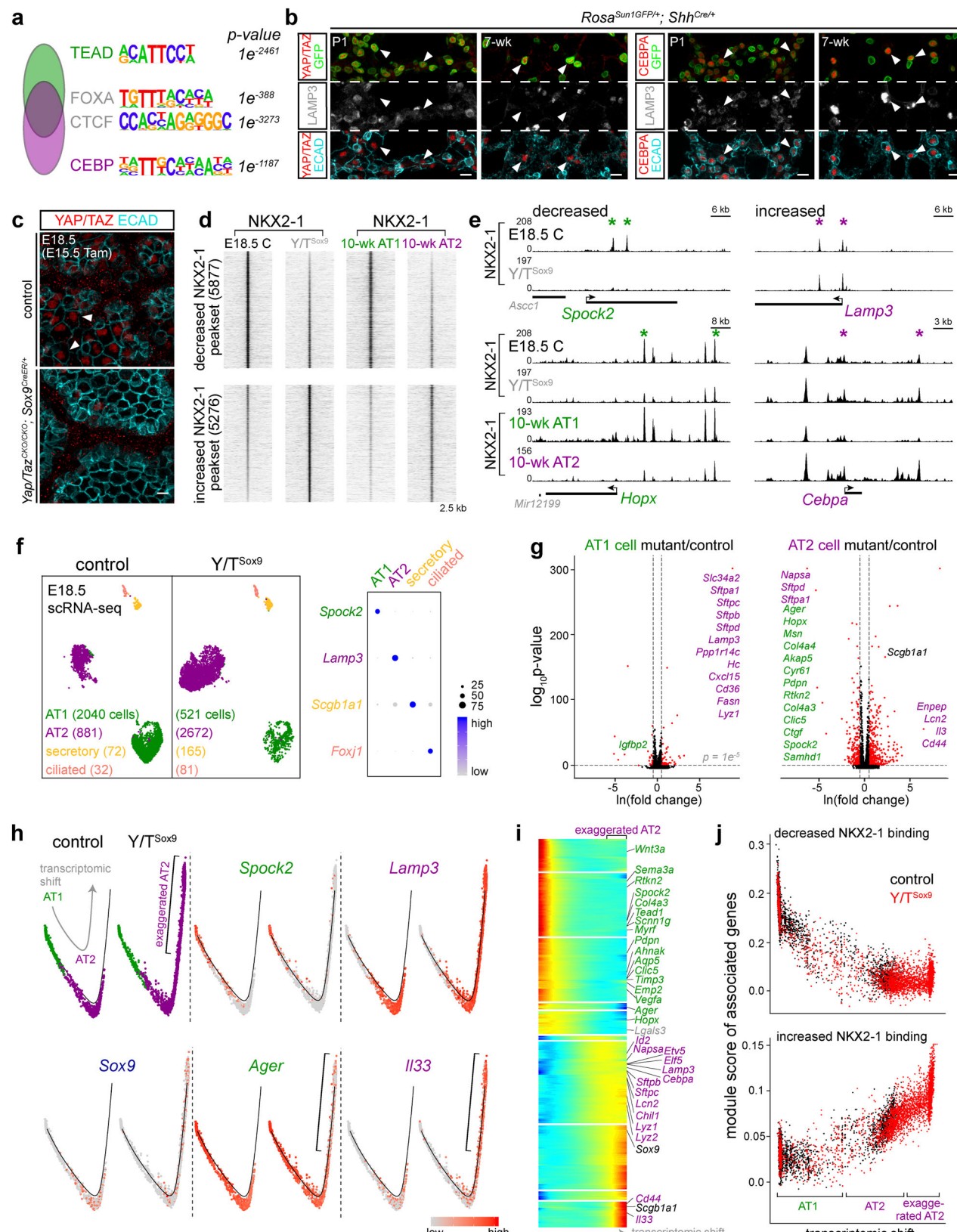

purified AT1 cells from P15 control and Y/T^Wnt3a mutant lungs (YAP/TAZ lost in 62% of 2147 GFP+ cells from 3 mice; Fig. 6a). Y/T^Wnt3a mutant AT1 cells had decreased NKX2-1 binding for AT1-specific sites and intriguingly again, increased binding for AT2-specific sites, as respectively exemplified by sites near *Spock2* and *Scnn1g* as well as *Lamp3* and *Cebpa* (Fig. 6b, c and

Supplementary Data 6), suggesting considerable plasticity of AT1 cells such that NKX2-1 relocated to AT2-specific sites in the absence of YAP/TAZ.

ScRNA-seq analysis showed that the Y/T^Wnt3a mutant had a cluster of AT1 cells that were transcriptionally distinct from their normal counterparts that presumably had escaped complete Cre

**Fig. 5 AT1-specific partner factors YAP/TAZ establish AT1-specific NKX2-1 binding and cell fate, and antagonize those of AT2 cells. a** Top enriched HOMER de novo motifs (binomial test) for AT1-specific (TEAD), common (FOXA and CTCF), AT2-specific (CEBP) NKX2-1 binding sites. **b** Nuclear YAP/TAZ and CEBPA are specifically detected (arrowhead) in AT1 and AT2 cells, respectively. $Shh^{Cre}$ genetically labels AT1 and AT2 nuclei, distinguishable by AT2 cell-specific LAMP3 and cuboidal ECAD. Scale: 10 μm. **c** Confocal images of immunostained E18.5 Y/T$^{Sox9}$ mutant and littermate control lungs received 3 mg tamoxifen (Tam) at E15.5, showing loss of nuclear YAP/TAZ from AT1 cells, as outlined by ECAD (arrowhead). Scale: 10 μm. **d** NKX2-1 ChIP-seq heatmaps of E18.5 Y/T$^{Sox9}$ mutant and littermate control (C) whole lungs, sorted by fold change and cross-referenced with NKX2-1 binding from Fig. 1. **e** Examples of NKX2-1 peaksets in **d**. *Lamp3* peaks do not reach statistical significance. NKX2-1 binding for *Spock2* and *Lamp3* in 10-week cells are shown in Fig. 1 (same in Fig. 6c). **f** ScRNA-seq UMAP comparison of Y/T$^{Sox9}$ mutant and littermate control epithelial cells with cell typing supported by the dot plot. The mutant has a decrease in the number of AT1 cells accompanied by an increase in the number of AT2 cells. **g** Volcano plots (MAST differential expression) comparing AT1 (left) and AT2 (right) cells in **f**. Differentially expressed, curated lists of AT1 and AT2-specific genes (Supplementary Data 6) are labeled (same in subsequent figures). **h** Monocle trajectory analysis of AT1 and AT2 cells in **f**, showing a linear transcriptomic shift from AT1 to AT2 cells in the control but further extending to transcriptionally exaggerated AT2 cells (bracket) in the Y/T$^{Sox9}$ mutant. The progenitor marker *Sox9* is present at a low level but normally higher in AT2 cells. **i** Monocle gene clusters with similar dynamics along the transcriptomic shift in **h**. **j** Seurat module scores of gene sets associated with the 20% most decreased (top) or increased (bottom) NKX2-1 binding sites in **d**, plotted along the Monocle trajectory in **h**, showing concordant changes in NKX2-1 binding and gene expression.

recombination (Fig. 6d). Comparison of AT1 cells in the control and mutant lungs revealed downregulation of AT1 genes and importantly, upregulation of AT2 genes, suggesting possible AT1-to-AT2 conversion (Fig. 6e, Supplementary Fig. 6, and Supplementary Data 7). This possibility was supported by a Monocle trajectory analysis that placed mutant AT1 cells between control AT1 and AT2 cells with intermediate levels of AT1 and AT2 genes (Fig. 6f, g and Supplementary Data 7). Remarkably, immunostaining showed that GFP-marked *Wnt3a^{Cre}*-lineage cells in the Y/T$^{Wnt3a}$ mutant expressed AT2 markers including SFTPC and LAMP3 with an increased frequency from P15 (19% of 1939 GFP+ cells from 3 mice) to 10-week-old (45% of 1833 GFP+ cells from 2 mice) and even became cuboidal as AT2 cells (Fig. 5h), providing genetic evidence for an unusual cell fate conversion of a terminally differentiated cell type. By design, AT2 cells were not targeted and thus were unaffected (Fig. 6e, h).

As in our analysis of the Y/T$^{Sox9}$ mutant, we assigned the 20% most decreased or increased NKX2-1 binding sites in the Y/T$^{Wnt3a}$ mutant to their nearest gene and derived an average expression score to correlate with the Monocle transcriptomic shift. The observed correlation supported the functionality of altered NKX2-1 binding (Fig. 6i and Supplementary Data 7). Intriguingly, the intermediate cells activated genes that were implicated in AT2-to-AT1 conversion during injury-repair, such as *Sfn*, *Krt8*, and *Lgals3*[36,37], suggesting a shared gene signature during cell fate changes (Fig. 6e–g and Supplementary Fig. 6a).

Notably, unsupervised principal component analysis of all our NKX2-1 binding datasets showed that the first component (PC-1; 58% of the variance) captured the temporal changes and the second component (PC-2; 17% of the variance) captured the differences among progenitor, AT1, and AT2 cells, recapitulating the gradual differentiation of E14.5 progenitors toward the opposing AT1 and AT2 cell fates (Fig. 6j). The E18.5 Y/T$^{Sox9}$ mutant drifted horizontally past AT2 cells, consistent with exaggerated AT2 cells; the P15 Y/T$^{Wnt3a}$ mutant AT1 cells drifted toward AT2 cells, consistent with AT1-to-AT2 conversion (Fig. 6j). Therefore, NKX2-1 binding over time and across mutants mirrored the corresponding transcriptomes (Figs. 3–6), supporting the concept that differential NKX2-1 binding resolves the AT1 and AT2 cell fates (Fig. 6k).

**Cell-type-specific *Nkx2-1* mutant cells explore distinct epigenetic space including the opposing cell fate.** The increased NKX2-1 binding to AT2-specific sites in both *Yap/Taz* mutants and associated shift of cells toward AT2 cell fate, together with the known AT2-to-AT1 differentiation during injury-repair[9,10], raised the possibility of a constant antagonism between the opposing AT1 versus AT2 cell fate. If true, we reasoned that loss

of one cell fate in our NKX2-1$^{Rtkn2}$ and NKX2-1$^{Sftpc}$ mutants might permit the opposing cell fate. Accordingly, we focused on sites with increased accessibility upon *Nkx2-1* deletion and found that those in the two *Nkx2-1* mutants had little overlap (445 sites shared between 18,367 and 11,275 sites) (Fig. 7a and Supplementary Data 8 and 9), suggesting that *Nkx2-1* mutant AT1 and AT2 cells underwent distinct epigenetic changes despite losing the same transcription factor.

In support of this, HOMER de novo motif analysis of newly accessible sites in the NKX2-1$^{Rtkn2}$ and NKX2-1$^{Sftpc}$ mutants revealed mostly unique motifs, except for the AP-1 motif that was shared in both mutants, perhaps reflecting a common stress response to cell cycle reentry or cell fate change[38] (Fig. 7b). Loss of NKX2-1 in either AT1 or AT2 cells were known to adopt an gastrointestinal fate during development, homeostasis, and tumorigenesis[11,39–42]. Indeed, 21 days after *Nkx2-1* deletion, both NKX2-1$^{Rtkn2}$ and NKX2-1$^{Sftpc}$ mutants formed aberrant epithelial cell clusters and expressed previously validated gastrointestinal markers PIGR and TFF2[11] (Fig. 7d). However, gastrointestinal genes shared between mutant AT1 and AT2 cells, such as *Tff2*, had gained little accessibility 5 days after *Nkx2-1* deletion (Supplementary Fig. 7a). There is evidence that FOXA1/A2 are released from NKX2-1 to activate a key gastrointestinal transcription factor, *Hnf4a*[39,40]; however, the FOXA motif was limited to the NKX2-1$^{Sftpc}$ mutant and *Hnf4a* had gained accessibility in both mutants, but at distinct sites (Fig. 7e), suggesting other mechanisms to activate gastrointestinal genes in the NKX2-1$^{Rtkn2}$ mutant (Fig. 7b). Interestingly, *Elf3* of the ELF motif family, a transcription factor required for gastrointestinal development[43], was ectopically expressed in *Nkx2-1* mutant AT1 cells (Supplementary Fig. 7b, c).

Intriguingly, one pair of motifs for the newly accessible sites were for the opposing cell fate: CEBP motif for the NKX2-1$^{Rtkn2}$ mutant and TEAD for the NKX2-1$^{Sftpc}$ mutant (Fig. 7b). When cross-referencing chromatin accessibility with NKX2-1 binding, we found that most increased sites were not associated with NKX2-1 binding, consistent with these sites being indirect targets of NKX2-1 (Fig. 7a and Supplementary Data 8 and 9)[11]. However, 10–15% of the sites had NKX2-1 binding albeit in the opposing cell type, and were more accessible in the opposing cell type of control lungs (Fig. 7a), suggesting increased accessibility in some AT2-specific genes in the NKX2-1$^{Rtkn2}$ mutant and some AT1-specific genes in the NKX2-1$^{Sftpc}$ mutant, as respectively exemplified by *Lyz2* and *Pdpn* (Fig. 7e). Despite increased accessibility, these genes were mostly not upregulated transcriptionally in previously published mutants[11] (Fig. 7c and Supplementary Data 10), suggesting that while *Nkx2-1* mutant AT1 and AT2 cells may explore epigenetic accessibility of the

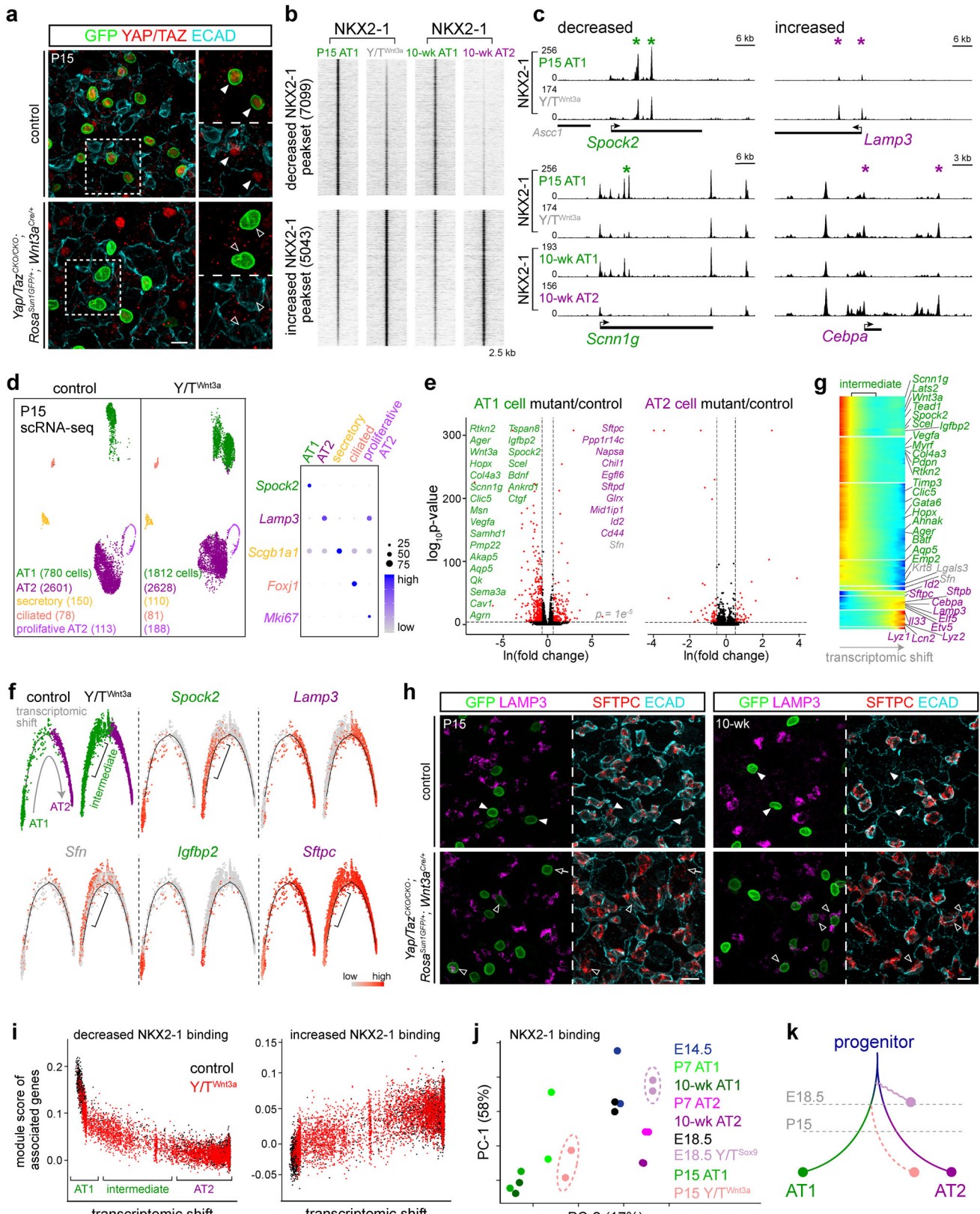

opposing cell fate, the new accessibility does not translate into gene expression, presumably due to the absence of NKX2-1. Taken together, the limited overlap between newly accessible sites and enriched motifs between the two *Nkx2-1* mutants suggested that mutant AT1 and AT2 cells converge onto the gastrointestinal

fate by following non-linear and distinct epigenetic paths including those toward the opposing cell fate (Fig. 7f). Future time-course analyses are necessary to track the epigenetic changes associated with the cell-type-specific shifts in cell fate and to identify the associated regulators.

**Fig. 6 YAP/TAZ maintain AT1-specific NKX2-1 binding and cell fate, and prevent AT1-to-AT2 conversion. a** Confocal images of immunostained lungs showing loss of YAP/TAZ in recombined AT1 cells (filled versus open arrowhead) in the Y/T^Wnt3a mutant. Scale: 10 μm. **b** NKX2-1 ChIP-seq heatmaps of purified AT1 nuclei from P15 Y/T^Wnt3a mutant and littermate control lungs, sorted by fold change and cross-referenced with NKX2-1 binding from Fig. 1. **c** Examples of NKX2-1 peaksets in **b**. **d** ScRNA-seq UMAP comparison of Y/T^Wnt3a mutant and littermate control epithelial cells with cell typing supported by the dot plot. AT1 cells in the mutant are higher in number, likely resulting from more efficient cell dissociation due to the phenotype in cell morphology (**h**). **e** Volcano plots (MAST differential expression) comparing AT1 (left) and AT2 (right) cells in **d**. **f** Monocle trajectory analysis of AT1 and AT2 cells in **d**, showing a linear transcriptomic shift from AT1 to AT2 cells in the control but via intermediate cells (bracket) in the mutant, which express a subset of AT1 (*Spock2*) and AT2 (*Sftpc*) genes as well as *Sfn*. **g** Monocle gene clusters with similar dynamics along the transcriptomic shift in **f**. **h** Confocal images of immunostained lungs showing AT2 markers (LAMP3 and SFTPC) in recombined cells in the Y/T^Wnt3a mutant (filled versus open arrowhead). Some SFTPC-expressing mutant cells still have extended morphology at P15 (open arrow), and become mostly cuboidal at 10 weeks. Scale: 10 μm. **i** Seurat module scores of gene sets associated with the 20% most decreased (left) or increased (right) NKX2-1 binding sites in **b**, plotted along the Monocle trajectory in **f**, showing concordant changes in NKX2-1 binding and gene expression. **j** Unsupervised principal component analysis of NKX2-1 binding across indicated color-coded samples. The E18.5 Y/T^Sox9 mutant is right-shifted as far as mature AT2 cells, whereas the P15 Y/T^Wnt3a mutant is between AT1 and AT2 cells, reflecting the exaggerated AT2 cells and intermediate cells in the respective mutants. **k** A diagram, reminiscent of and color-coded as in **j**, depicting the normal differentiation of progenitors toward AT1 and AT2 cells, as well as the drift in both NKX2-1 binding and transcriptome observed in Y/T^Sox9 and Y/T^Wnt3a mutants.

## Discussion

Our native tissue-derived genomic data have delineated the in vivo function of the lung lineage transcription factor NKX2-1 in opposing cell types and across developmental stages. Key unexpected findings include (1) cell-type-specific NKX2-1 binding is preferentially acquired and, surprisingly, retained as progenitors differentiate into each cell type, supporting the concept of cell fate and potential marked by NKX2-1 binding; (2) the AT1 and AT2 cell fates continuously antagonize each other so that YAP/TAZ and by extension TEADs function as partner factors of NKX2-1 in AT1 cells, restricting NKX2-1 binding to AT1-specific sites and preventing cell fate conversion in a development stage-dependent manner; (3) loss of NKX2-1 allows AT1 and AT2 cells to gain epigenetic features of AT2 and AT1 cells, respectively, without corresponding transcriptional activation, suggesting that a lineage transcription factor can be coerced in one cell type to inhibit the epigenetic state of the opposing cell type. Our cell-type-specific epigenomic and genetic study sheds light on the molecular logic of resolving opposing cell fates by a lineage transcription factor in native tissues.

Lineage transcription factors including NKX2-1 mark a given tissue lineage and by definition are transcriptionally equivalent among cell types within the lineage. However, on the protein level, they could theoretically have cell-type-specific posttranslational modifications, DNA-binding targets, or transcriptional cofactors—possibilities that are discernible only by comparing pure cell type populations and practically often addressed in cultured cells[6–8]. The abundance of AT1 and AT2 cells and their robust genetic drivers allow us to identify shared and distinct NKX2-1 binding sites and test the functionality of such binding in native tissues. We show that cell-type-specific NKX2-1 binding sites, compared to common ones, are more often associated with distal regulatory elements and are functional in regulating chromatin accessibility, extending such general transcriptional mechanisms[6] to opposing cell types of the same lineage in vivo. Integrating scATAC-seq data that can now be readily obtained for native tissues, we parse the aforementioned common NKX2-1 binding sites into lineage and housekeeping ones, the former of which lose accessibility upon *Nkx2-1* deletion (Figs. 1 and 2), supporting a shared function of NKX2-1 in AT1, AT2 and possibly airway cells. By comparison, NKX2-1 binding to housekeeping sites including possible insulators implies its more general role in chromatin organization, although site accessibility is largely unaffected without NKX2-1, possibly due to redundancy with other transcription factors expected at these generic sites. Future 3D chromatin analysis[44] and extension of our approaches to airway cell types as well as other tissues will provide a complete picture of how lineage transcription factors function in vivo.

Cell-type-specific NKX2-1 binding cannot be explained by the bound DNA sequences alone as they are identical in AT1 and AT2 cells. Nevertheless, motif analysis of the NKX2-1-bound sequences in AT1 versus AT2 cells identifies the expected shared NKX motif, as well as AT1-specific TEAD motif and AT2-specific CEBP motif, consistent with binding specificity as a result of partner transcription factors. Indeed, AT1-specific NKX2-1 binding depends on YAP/TAZ, cofactors of TEADs (Figs. 5 and 6)—prompting future investigation of the role of CEBPs in AT2-specific NKX2-1 binding beyond existing phenotypic characterization[45], as well as equivalent partner factors in NKX2-1-expressing airway cells or in the context of injury-repair or tumorigenesis using more sensitive variants of ChIP-seq[46]. Activatable by cell stretching[31] possibly from lung growth and/or inspiration, YAP/TAZ could recruit NKX2-1 to promote AT1 cell growth, which in turn releases tension and prevents additional cells from adopting the AT1 cell fate—a negative feedback mechanism to generate a mosaic of AT1 and AT2 cells that is distinct from Notch-mediated lateral inhibition and possibly allows continuous antagonism between AT1 and AT2 cell fates. In support of this, loss of YAP/TAZ shifts NKX2-1 to its AT2-specific sites, eventually leading to AT2 gene expression and morphology (Fig. 6). We note that, although NKX2-1 and YAP/TAZ/TEAD can exist in a complex in vitro[33,34], a limitation of this study to be addressed by future experiments is to examine possible biochemical interactions between NKX2-1 and its partner factors in purified cell types from native tissues.

The conversion among AT1, AT2, and gastrointestinal fates in this study and the literature highlights remarkable cellular plasticity, even for a terminally differentiated cell type such as the AT1 cell[9–11,24,40–42,47–49]. The theoretical potential of a cell is only limited by its DNA sequence, as demonstrated in the extreme case of inducing pluripotent stem cells from fibroblasts[50]. However, during normal development and homeostasis, the physiological potential of a cell is much more limited, as conceptualized in the Waddington landscape model[51]—testable with lineage-tracing and transplant experiments—and exemplified, in this study, by the potential of SOX9 progenitors to form both AT1 and AT2 cells. We show that at least one underlying mechanism is NKX2-1 binding to retained sites, marking them for future expression (Figs. 3 and 4). More often, the literature illustrates the experimental potential of a cell, where loss or gain-of-function manipulations alter a cell fate. Loss-of-function settings often reveal a potential based on ongoing antagonism or shared developmental origin, as exemplified in the AT1–AT2 balance or the endodermal origin of the lung and the gut, respectively, and demonstrated in our *Yap/Taz* and *Nkx2-1* mutants (Figs. 5–7). Gain-of-function settings including directed differentiation are widely used in regenerative medicine but often

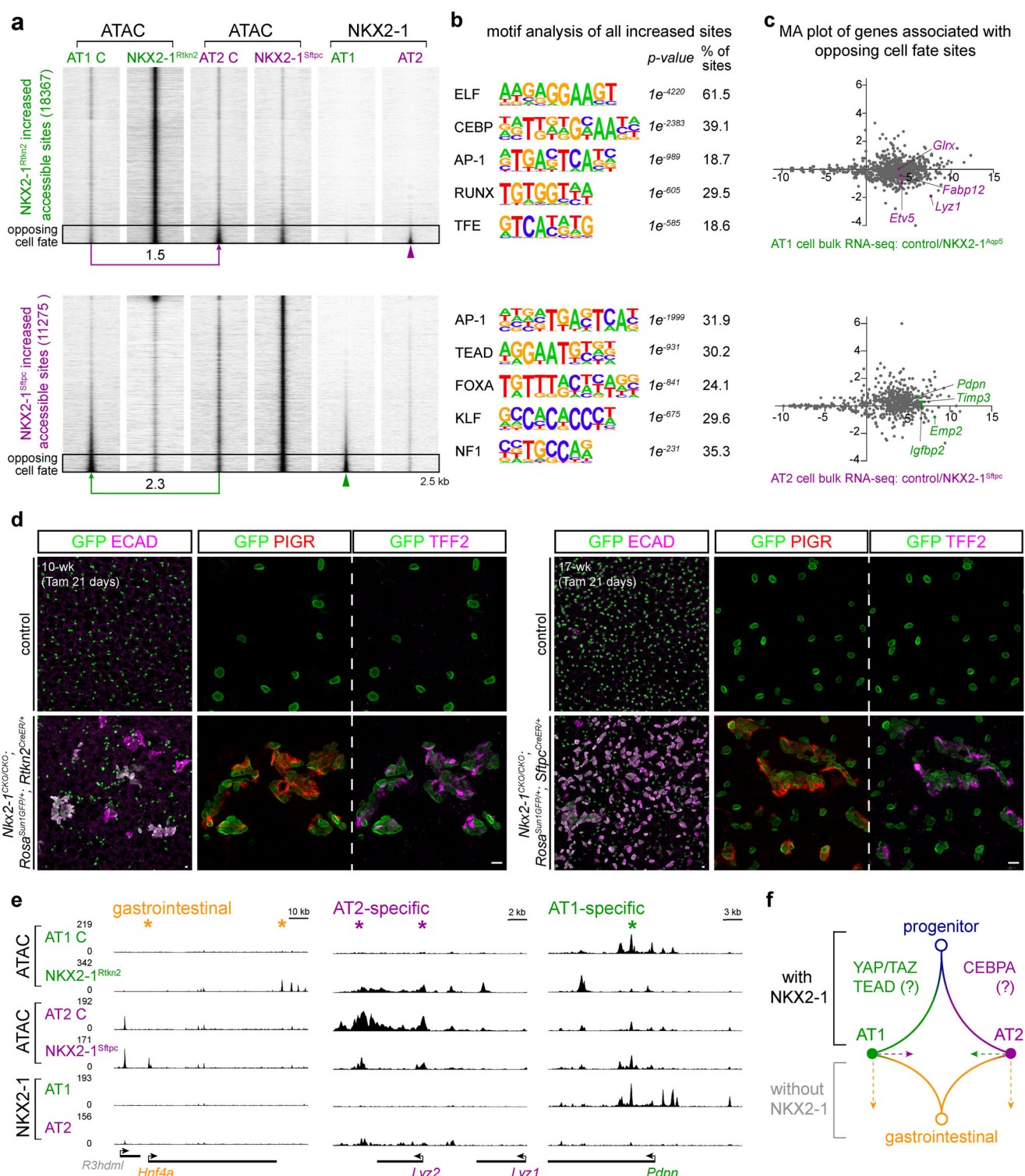

do not fully recapitulate the intended cell fate[52], possibly due to the ability of cells to explore a larger epigenetic space (Fig. 7) in addition to the difficulty in precisely controlling the level and duration of the overexpression. Systematic comparison of the epigenome and transcriptome underlying the theoretical, physiological, and experimental potentials will supplement ongoing efforts in cataloging all cell types in the body[53]. These cellular potentials could be unleashed under pathological conditions. For example, NKX2-1 is considered a tumor suppressor in lung cancer—a function perhaps attributable to its role in limiting

cellular potentials such that the increased epigenetic plasticity of NKX2-1 mutant tumor cells may allow adaptation and growth advantage in the tumor microenvironment[49,54]. Furthermore, the observed proliferation upon NKX2-1 loss could provide the substrate for or synergize with additional oncogenes and tumor suppressors.

## Methods

**Mice *Mus Musculus*.** The following mouse strains were used: Nkx2-1[CKO55], Yap[CKO56], Taz[CKO56], Wnt3a[Cre17], Sftpc[CreER18], Sox9[CreER57], Shh[Cre58], Rosa[Sun1GFP16],

**Fig. 7 Cell-type-specific *Nkx2-1* mutant cells explore distinct epigenetic space including the opposing cell fate. a** ATAC-seq heatmaps of sites with increased accessibility in purified AT1 (top) and AT2 (bottom) cells from the respective NKX2-1$^{Rtkn2}$ and NKX2-1$^{Sftpc}$ mutants as in Fig. 2, cross-compared with and sorted by NKX2-1 binding in 10-week-old AT1 versus AT2 cells from Fig. 1. The sites from the two mutants have limited overlap and are largely not bound by NKX2-1, although 10% sites (boxed area) have NKX2-1 binding (arrowhead) and corresponding increased accessibility (log2 fold change shown) in control cells of the opposing fate. **b** Top enriched HOMER de novo motifs (binomial test) for sites in **a** as well as their *p* values and percentage of sites containing the predict motif. Note limited overlap except for the AP-1 motif and CEBP and TEAD motifs in the opposing cell types. **c** MA plots of genes associated with the boxed 10% sites in **a** using published bulk RNA-seq data of AT1-specific and AT2-specific *Nkx2-1* mutant developing lungs[11], showing that increased accessibility of sites of the opposing cell fate does not correspond to gene upregulation. **d** Confocal images of immunostained mature lungs from AT1-specific (left panels) and AT2-specific (right panels) *Nkx2-1* mutants and their respective littermate controls, showing aberrant epithelial cell clusters and expression of gastrointestinal markers PIGR and TFF2 in both mutants. Two doses (3 mg each; 48 h interval) of tamoxifen (Tam) were given and the lungs were harvested at 21 days after the initial dose. Scale: 10 µm. **e** Example increased accessibility in sites near a gastrointestinal gene (*Hnf4a*; distinct sites for NKX2-1$^{Rtkn2}$ versus NKX2-1$^{Sftpc}$ mutants and annotated to a nearby gene *Ttpal*), AT2-specific NKX2-1 sites (*Lyz2*) in the NKX2-1$^{Rtkn2}$ mutant, and AT1-specific NKX2-1 site (*Pdpn*) in the NKX2-1$^{Sftpc}$ mutant. The decreased accessibility of *Lyz2* and *Pdpn* sites in concordant mutants are examined in Fig. 2. **f** A diagram to depict that NKX2-1 promotes the differentiation of progenitors toward AT1 or AT2 fate possibly in partner with YAP/TAZ/TEAD or CEBPA, respectively. Without NKX2-1, AT1 and AT2 cells drift toward the opposing cell fate while adopting the gastrointestinal fate.

---

*Rosa$^{mTmG59}$*, and *Rtkn2$^{CreER}$* (this study). The *Rtkn2$^{CreER}$* knock-in allele was generated using CRISPR targeting via standard pronuclear injection[60]. Specifically, 400 nM gRNA (Synthego), 200 nM Cas9 protein (E120020-250ug, Sigma), and 500 nM circular donor plasmid were mixed in the injection buffer (10 mM Tris pH 7.5, 0.1 mM EDTA). The 5′ homology arm in the donor plasmid was PCR amplified between 5′-CCACTTGGATCCTGGGGATTGGAA and 5′-GATTTGAAAAGCGCG CCCCAGGGC; the 3′ homology arm was PCR amplified between 5′-AGGGGCAG CTGCTGAGGGGTCTCG and 5′-CTTAACAGATCTCCATTTAGTTCA. The gRNA targeted 5′-GGCCGTGCCTTGCACCGAGATGG with the last three nucleotides being the protospacer adjacent motif (PAM; not included in gRNA) and the start codon underlined and replaced by that of CreER used in *Sox9$^{CreER57}$*. Uncropped images of *Rtkn2$^{CreER}$* genotyping gels are provided in the Source Data file.

Observation of a vaginal plug was designated as E0.5 and intraperitoneal injections of tamoxifen (T5649, Sigma) dissolved in corn oil (C8267, Sigma) at doses specified in figure legends were performed to activate the Cre recombinase. The mice used were of both gender and mixed genetic background, and experiments were carried out with investigators not blind to the genotypes. The mice were housed under conditions of 22 °C, 45% humidity, and 12–12 h light–dark cycle. To reduce experimental variation, samples were processed in the same tissue blocks or tubes. Sample sizes for experiments were not determined by power analysis. All animal experiments were approved by the Institutional Animal Care and Use at MD Anderson Cancer Center.

**Antibodies.** The following antibodies were used for immunofluorescence: rat anti-protein tyrosine phosphatase, receptor type, C (CD45, 1:2000, 14-0451-81, eBioscience) rabbit anti-CCAAT/enhancer binding protein alpha (C/EBPA, 1:500, 8178P, Cell Signaling Technology), rat anti-epithelial cadherin (ECAD, 1:1000, 13190, Life Technology), chicken anti-green fluorescent protein (GFP, 1:5000, AB13970, Abcam), rabbit anti-homeodomain only protein (HOPX, 1:500, sc-30216, Santa Cruz), rat anti-KI67 (KI67, 1:1000, 14-5698-82, Invitrogen), guinea pig anti-lysosomal associated membrane protein 3 (LAMP3, 1:500, 391005, SySy), rabbit anti-NK homeobox 2-1 (NKX2-1, 1:1000, sc-13040, Santa Cruz), goat anti-podoplanin (PDPN, 1:1000, AF3244, R&D), goat anti-polymeric immunoglobulin receptor (PIGR, 1:1000, AF2800, R&D), rabbit anti-pro-surfactant protein C (SFTPC, 1:1000, AB3786, Millipore), rabbit anti-trefoil factor 2 (TFF2, 1:1000, 13681-1-AP, ProteinTech), rabbit anti-Yes-associated protein 1 and WW domain containing transcription regulator 1 (YAP1 and WWTR1/TAZ, 1:250, D24E4, Cell Signaling Technology). The following antibodies were used for fluorescence activated cell sorting: PE/Cy7 rat anti-CD45 (CD45, 1:250, 103114, BioLegend), PE rat anti-epithelial cadherin (ECAD, 1:250, 147304, BioLegend), BV421 rat anti-epithelial cell adhesion molecule (EPCAM, 1:250, 118225, BioLegend), and AF647 rat anti-Intercellular adhesion molecule 2 (ICAM2, 1:250, A15452, Thermo Fisher). The following antibodies were used for chromatin immunoprecipitation: rabbit anti-histone H3 lysine 27 acetylation (H3K27ac, 1 µg/ml, ab4729, Abcam), rabbit anti-Histone H3 lysine 4 mono-methylation (H3K4me1, 0.6 µg/ml, ab8895, Abcam), rabbit anti-Histone H3 lysine 4 tri-methylation (H3K4me3, 1 µg/ml, ab8580, Abcam), and rabbit anti-NK Homeobox 2-1 (NKX2-1, 1 µg/ml, ab133737, Abcam).

**Harvesting lungs for immunostaining.** Lungs were inflated using a gravity drip as published previously with minor modifications[11]. Avertin (T48402, Sigma) was used to anaesthetize mice. The right ventricle of the heart was then injected with phosphate buffered saline (PBS, pH 7.4) to perfuse the lung. Inflation was achieved through 25 cm H₂O pressure gravity drip of a 0.5% paraformaldehyde (PFA, P6148, Sigma) in PBS through a cannulated trachea. Inflated lungs were then fixed by submersion in 0.5% PFA for 3–6 h on a rocker at room temperature then washed overnight at 4 °C on a rocker. Lobes were then dissected with wholemount strips cut or transferred to a 20% sucrose in PBS solution containing 10% optimal cutting temperature compound (OCT, 4583, Tissue-Tek) to cryoprotect samples. Samples were then incubated overnight at 4 °C on a rocker and frozen the next day in OCT.

**Section immunostaining.** Samples embedded as described above were cryosectioned at 10 or 20 µm thickness. After air drying for 1 h, sections were blocked in 5% normal donkey serum (017-000-121, Jackson ImmunoResearch) and PBS with 0.3% Triton X-100 (PBST). Primary antibodies were diluted in PBST and added for incubation in a humidified chamber at 4 °C overnight. Sections were then washed 30 min in a coplin jar with PBS followed by incubation with 1:1000 diluted secondary antibodies (Jackson ImmunoResearch) and 4′,6′-diamidino-2-phenylindole DAPI (D9542, Sigma) if applicable for 1 h at room temperature. After a second 30 min wash with PBS, samples were mounted using Aquapolymount (18606, Polysciences) and imaged either using a Nikon A1 plus confocal microscope or an Olympus FV1000 confocal microscope.

**Wholemount immunostaining.** Previously published protocols for wholemount immunostaining were followed with minor modifications[11]. From the cranial or left lobes of lungs, 3 mm thick strips were cut or 60 µm cryosections were collected for staining. A solution of 5% normal donkey serum in PBST was used to block samples at room temperature on a rocker for 2 h. Primary antibodies diluted with PBST on a rocker at 4 °C overnight. Samples were then washed three times over 3 h with PBS + 1% Triton X-100 + 1% Tween-20 (PBSTT) followed by incubation on a rocker at 4 °C overnight with donkey secondary antibodies and DAPI diluted in PBST (1:1000). Strips were then washed again three times over the 3 h with PBSTT then incubated with 2% PFA in PBS for 2–3 h on a rocker at room temperature. Samples were then mounted on Premium Beveled Edge microscope slides (8201, Premiere) with flat side facing the coverslip using Aquapolymount between electrical tape to relieve pressure from the coverslip that could deform the tissue. After drying, samples were then imaged using a Nikon A1 plus confocal microscope or an Olympus FV1000 confocal microscope.

**Immunostaining quantifications.** Quantification of driver specificity, efficiency, and deletion within the alveolar epithelium was carried out using confocal images of GFP, NKX2-1, ECAD, and LAMP3 stained lungs that were taken either with a ×40 oil objective or ×20 oil objective on wholemount immunostained strips (318 × 318 × 20 µm for *Wnt3a$^{Cre}$* at 10-week-old and P7 and *Sftpc$^{CreER}$* at P7) or sections (636 × 636 × 10 µm for *Rtkn2$^{CreER}$*, NKX2-1$^{Rtkn2}$, *Sftpc$^{CreER}$*, and NKX2-1$^{Sftpc}$). Quantification of cell proliferation was carried out on confocal images of GFP, NKX2-1, and KI67 stained lungs that were taken with ×20 oil objective on immunostained sections (636 × 636 × 10 µm for NKX2-1$^{Rtkn2}$ and NKX2-1$^{Sftpc}$). Efficiency of deletion in Y/T$^{Wnt3a}$ was quantified using confocal images of GFP, YAP/TAZ, and CD45 stained lungs that were taken with a ×20 oil objective on wholemount immunostained strips (635 × 635 × 20 µm). The percentage of YAP/TAZ positive cells that were CD45 negative and GFP positive was used to calculate deletion. Quantification of SFTPC in Y/T$^{Wnt3a}$ was carried out on confocal images of GFP, SFTPC, and ECAD stained lungs (636 × 636 × 10 µm; Y/T$^{Wnt3a}$) taken with a ×20 oil objective on immunostained sections; GFP+ cells were considered positive if they were encircled by SFTPC. For each set of immunostaining, the quantifications were performed on either the full image or a random half of the image and cells were manually categorized.

**Cell dissociation and sorting cells.** Perfused lungs were collected from anesthetized mice as described above. Previously published protocols for cell dissociation and sorting cells were used with minor modifications[11]. Extra-pulmonary airways and connective tissues were removed from lungs, which were then minced with forceps and digested in Liebovitz media (Gibco, 21083-027) for scRNA-seq,

ATAC-seq, or scATAC-seq with 2 mg/ml collagenase type I (Worthington, CLS-1, LS004197), 0.5 mg/ml DNase I (Worthington, D, LS002007), and 2 mg/ml elastase (Worthingon, ESL, LS002294) for 30 min at 37 °C. To stop the digestion, fetal bovine serum (FBS, Invitrogen, 10082-139) was added to a final concentration of 20%. Tissues were triturated until homogenous and filtered through a 70 μm cell strainer (Falcon, 352350) on ice in a 4 °C cold room and transferred to a 2 ml tube. The sample was then centrifuged at 1537 rcf for 1 min; then 1 ml red blood cell lysis buffer (15 mM $NH_4Cl$, 12 mM $NaHCO_3$, 0.1 mM EDTA, pH 8.0) was added and incubated for 3 min before centrifugation again at 1537 rcf. The red blood cell lysis buffer incubation was repeated to remove residual red blood cells. Liebovitz + 10% FBS was used to wash and resuspend samples, which were then filtrated through a 35 μm cell strainer into a 5 ml glass tube and had SYTOX Blue (1:1000, Invitrogen, S34857) added. After refiltering, samples were for ATAC-seq were sorted for GFP+ cells from a $Rosa^{Sun1GFP}$ reporter activated by $Wnt3a^{Cre}$, $Rtkn2^{CreER}$, or $Sftpc^{CreER}$ on a BD FACSAria IIIu cell sorter with a 70 μm nozzle. Samples for scRNA-seq or scATAC-seq were stained with CD45-PE/Cy7 (BioLegend, 103114), ECAD-PE (BioLegend, 147304), and ICAM2-A647 (Invitrogen, A15452) antibodies (1:250 dilutions for all antibodies) for 30 min followed by a wash and being resuspended with Liebovitz meda + 10% FBS and addition of SYTOX Blue. Samples were then filtered through a 35 μm cell strainer into a 5 ml glass tube again and sorted by a BD FACSAria Fusion sorter or a BD FACSAria IIIu cell sorter with a 70 μm nozzle. Cell sorting data were analyzed using FlowJo 9.0.

**Single-cell RNA-seq.** Cells sorted as described above were processed using the 3′ Library and Gel Bead Kit following the manufacturer's users guide (v2 rev D) on the Chromium Single-Cell Gene Expression Solution Platform (10X Genomics). The scRNA-seq for $Sox9^{CreER/+}$; $Yap1^{CKO/CKO}$; $Taz^{CKO/CKO}$ and control sample were processed using the 3′ Library and Gel Bead Kit following the manufacturer's users guide (v3 rev D). All libraries were then sequenced using Illumina NextSeq500 or Novaseq6000 with a 26 × 124 format with 8 bp index (Read1). Each sample group, such as control versus mutant or 12-sample controls, was merged using Cell Ranger's "cellranger count" and "cellranger aggr", which struck a balance between ameliorating batch effects and preserving biological differences. Data from downstream analysis were performed using Seurat R package (v3)[28] and custom R scripts. Cells with gene counts over 5000 or less than 200 were filtered out. Immune, mesenchymal, endothelial, and epithelial lineage clusters were identified using Ptprc, Col3a1, Icam2, and Cdh1 as previously published[11]. Epithelial lineages were then subset for further clustering. After reclustering epithelial cells, the lineage marks were checked again and if full clusters had expression of the aforementioned lineage associated genes, they were considered doublets and removed from the subsequent analyses. Epithelial cell types were identified using Foxj1 for ciliated cells, Scgb1a1 for secretory cells, Trp63 for basal cells, Ascl1 for neuroendocrine cells, Spock2 for AT1 cells, Lamp3 for AT2 cells, Sox2 for airway progenitors at E14.5, and Sox9 for alveolar progenitors at E14.5. SOX9 progenitors, AT1 cells, and AT2 cells were then subset for further analysis. Module scores were calculated using Seurat for gene sets from ChIP-seq analyses. Model-based Analysis of Single-cell Transcriptomics (MAST) was used to identify differentially expressed genes[61]. Control and mutant lungs were processed in parallel experimentally and computationally, and thus spatially comparable in the UMAP plots.

**Pseudotemporal single-cell RNA-seq data analysis.** Monocle 2.8.0 was used to analyze Seurat clusters in either of progenitor, AT1, and AT2 cells from aggregate control and mutant samples for $Sox9^{CreER/+}$; $Yap1^{CKO/CKO}$; $Taz^{CKO/CKO}$ and $Wnt3a^{Cre/+}$; $Yap1^{CKO/CKO}$; $Taz^{CKO/CKO}$ or 12 aggregated control samples from E14.5, E16.5, E18.5, P4, P6, P7, P8, P10, P15, P20, 10-week-old, and 15-week-old lungs[11,20,23,62-65]. Cells were ordered using the top 2000 variable genes for control and mutant cells, and 1000 genes for the 12-aggregate sample identified by Seurat to generate pseudotime trajectories. Pseudotime for control and mutant lung cells, referred to as transcriptomic shift, was exported into a comma separate file along with the module scores calculated with each cell. The AT2 pseudotime trajectory was adjusted to have the same endpoint as that of the AT1 pseudotime trajectory, as expected for cells from the same lung. Monocle heatmaps were generated using the 1000 most differential genes across pseudotime.

**Bulk RNA-seq.** The previously published RNA-seq datasets[11] for deletion of Nkx2-1 in AT1 and AT2 cells during development were used to analyze expression of genes associated with 10% of sites (boxed in Fig. 7a) with increased accessibility in NKX2-1$^{Rtkn2}$ and NKX2-1$^{Sftpc}$ mutants.

**Tissue dissociation and sorting nuclei.** Lungs were harvested from Avertin anesthetized mice after perfusing 3 ml of cold PBS through the right ventricle. Lungs were minced after the removal of extra-pulmonary tissues. The tissue was then crosslinked for 20 min on a rocker at room temperature using a 1:4 PBS diluted 10% buffered formalin (Thermo Fisher Scientific, 23-245-685). To quench the excess fixative, 1 M glycine (pH 5.0) was added to a final concentration of 125 mM and incubated at room temperature on a rocker for 5–10 min. The fixed tissue was then washed with 2 ml cold PBS and resuspended with 1 ml (500 μl for embryos) of isolation of nuclei tagged in specific cell types (INTACT[16]) buffer (20 mM HEPES

pH 7.4, 25 mM KCl, 0.5 mM MgCl2, 0.25 M sucrose, 1 mM DTT, 0.4% NP-40, 0.5 mM Spermine, 0.5 mM Spermidine) with protease inhibitor cocktail (cOmplete ULTRA Tablets, Mini, EDTA-free, EASY pack, Sigma, 5892791001 or Pierce Protease Inhibitor Mini Tablets, EDTA-free, Thermo Fisher Scientific, A32955). Resuspended samples were then Dounce homogenized for 5 strokes, filtered through a 70 μm cell strainer, and centrifuged in a 2 ml tube at 384 rcf for 5 min. Samples were then resuspended in PBS plus protease inhibitor cocktail and either sorted for cell-type-specific ChIP-seq or counted for whole lung ChIP-seq. For cell-type-specific ChIP-seq, nuclei were stained with Sytox blue (1:1000, Invitrogen, S34857) then filtered through a 35 μm cell strainer into a 5 ml glass tube (12 × 75 mm Culture Tubes with closures volume 5 ml, VWR, 60818-565) blocked with 200 μL 10 mg/ml BSA (Sigma, A3059) and 1x protease inhibitor cocktail. Nuclei were then sorted at 4 °C using a BD FACSAria Fusion sorter or BD FACSAria IIIu cell sorter for GFP+ nuclei from the $Rosa^{Sun1GFP}$ allele and Sytox blue positive nuclei into a 1.7 ml collection tube containing and blocked with 300 μl of 10 mg/ml BSA with 5x protease inhibitor cocktail. $Wnt3a^{Cre/+}$; $Rosa^{Sun1GFP/+}$ mice rendered ~1 million nuclei per lung. $Sftpc^{CreER/+}$; $Rosa^{Sun1GFP/+}$ mice rendered 1–2 million nuclei per lung. For a full set of histones ChIPs in addition to an NKX2-1 ChIP, nuclei from lungs of mice with the same genotype were combined. If samples were combined, a second experiment with different mice of the same genotype and time point would be performed for a biological replicate.

**Chromatin immunoprecipitation.** The published method for chromatin immunopreciation was used with minor modifications[11]. Whole lung samples or sorted nuclei were split into aliquots of one million nuclei per 1.7 ml tube. Nuclei were then centrifuged at 12,052 rcf for 10 min at 4 °C. The supernatant was discarded and the visible pellet in each aliquot was then resuspended in 100 μl of ChIP nuclei lysis buffer (50 mM Tris-HCl pH 8.1, 10 mM EDTA, 1% SDS with 1x protease inhibitor cocktail) and incubated for 15 min at 4 °C. Concurrently, two sets of Protein G Dynabeads (Thermo Fisher Scientific, 10004D) were blocked with 200 μl 20 mg/ml bovine serum albumin (Jackson ImmunoResearch, 001-000-161), 4 μl 10 mg/ml salmon sperm DNA (Invitrogen, 15632-011), and ChIP dilution buffer (16.7 mM Tris-HCl pH 8.1, 1.2 mM EDTA, 1.1% Triton X-100, 0.01% SDS with 1× protease inhibitor cocktail). The first set of beads consisted of 40 μl of Protein G Dynabeads per sample was blocked for 1 h on a rotator for use in preclearing the chromatin. The second set with 30 μl protein G Dynabeads per antibody for ChIP was blocked on a rotator overnight a 4 °C. Nuclei were sonicated at 4 °C using a Bioruptor Twin (Diagenode, UCD-400-TO) for 38 cycles of 30 s on 30 s off on the high setting for a target DNA fragment size of 200–500 bp. Samples were then pooled if they originated from the same sample to make one set of antibodies (NKX2-1, H3K4me3, H3K4me1, H3k27ac, and H3K27me3) for a genotype, which would be considered one replicate for the respective antibodies. Samples were then centrifuged at 4 °C at 12,052 rcf for 15 min. Concurrently, the first set of blocked Protein G Dynabeads was washed twice with ChIP dilution buffer using the magnetic adapter and transferred to a fresh 2 ml tube. Two 20 μl inputs were added to 300 μl of Tris-EDTA (TE) buffer (10 mM Tris-HCl pH 8.1, 1 mM EDTA) and stored overnight at −80 °C. The remaining chromatin was added to the washed blocked Protein G Dynabeads, diluted to 1 ml with ChIP dilution buffer, and precleared on a rotator for 1 h at 4 °C. With a magnetic adapter, chromatin was split into fresh 2 ml tubes for incubation with antibodies overnight and diluted to 1 ml with ChIP dilution buffer as necessary. The next morning, the second set of blocked Protein G Dynabeads was washed using ChIP dilution buffer twice and transferred to fresh 2 ml tubes, to which the antibody-chromatin solution was added. This solution was incubated at 4 °C while rotating for 3 h. Using a magnetic adapter, the beads were washed with 1 ml of each of the following prechilled buffers until the beads were completely resuspended: low salt buffer (150 mM NaCl, 2 mM EDTA, 1% Triton X-100, 20 mM Tris-HCl pH 8.1, 0.1% SDS), high salt buffer (500 mM NaCl, 2 mM EDTA, 1% Triton X-100, 20 mM Tris-HCl pH 8.1, 0.1% SDS), lithium chloride buffer (250 mM LiCl, 1 mM EDTA, 1% NP-40, 10 mM Tris-HCl pH 8.1, 0.1% sodium deoxycholate), and TE buffer twice. The second wash of TE buffer was used to transfer the beads to a fresh 2 ml tube, which were resuspended with 300 μl of TE buffer. The frozen inputs were then thawed and incubated along with the samples with 1.5 μl of 10 mg/ml RNase A (Qiagen, 1007885) for 1 h at 37 °C. Samples were then switched to a 55 °C incubator for 4 h after addition of 15 μl 10% Sodium dodecyl sulfate and 3.5 μl 20 mg/ml Proteinase K (Thermo Fisher, EO0491). Then reverse crosslinking was achieved by 65 °C overnight incubation. The next day, 320 μl of phenol:chloroform:isoamyl alcohol solution (Sigma, P2069-400ML) was added to samples and input, which was followed by centrifugation at 12,052 rcf for 15 min at 4 °C for DNA extraction. The upper phase containing DNA was transferred to a new tube and precipitated with 2 volumes of 100% ethanol, 1/10 volume of 3 M NaCl, and 3 μl of 20 μg/μl glycogen (Invitrogen, 10814-010) followed by a brief vortex and centrifugation for 30 min at 12,052 rcf at 4 °C. DNA pellets were washed with 70% ethanol and centrifuged at 12,052 rcf at 4 °C. The supernatant was discarded and pellets were allowed to air dry for 10 min and then dissolved in 10 μl nuclease-free $H_2O$.

**ChIP-seq library preparation.** Qubit dsDNA HS Assay Kit (Invitrogen, Q23851) was used to measure the quantity of ChIP DNA. Then <5 ng ChIP sample DNA or <20 ng input DNA was used for sequencing libraries using the NEB Next Ultra II

DNA Library Prep Kit for Illumina (New England BioLabs, E7645). Two-step purification was reduced to one step by skipping step 3.1 per manufacturer's recommendation to improve the quality of the final output library. The DNA library for each sample was PCR amplified for 12 cycles using indexed primers (New England BioLabs, E7335S or E7500S) to barcode samples. Sample products then underwent a double-sided (0.65 × −1 × volume) size selection and purification using SPRIselect magnetic beads (Beckman Coulter, B23318). Concentrations were measured using the Qubit HS dsDNA assay then the size and purification of primer dimmers of DNA libraries was verified by gel electrophoresis. Samples were then combined with less than 20 barcoded samples per sequencing run on an Illumina NextSeq500.

**ChIP-seq analysis**. Reads of the same barcode were combined and the quality of reads were assessed using FastQC (http://www.bioinformatics.babraham.ac.uk/projects/fastqc/). Trimming of extra base paired of poor quality on the ends of reads was carried out using Trimmomatic[66] and was followed by another round of FastQC to assess read quality. High quality reads were then aligned using Bowtie[67,68] with the following settings: -m1 –k1 –v1. After alignment, sam files were converted to bam files and filtered for unmapped reads, chimeric alignments, low quality alignments, and PCR duplicates using Picard[69] MarkDuplicates and samtools[70] settings: -b -h -F 4 -F 1024 -F 2048 -q 30. SPP cross correlation[71] was carried out and all samples passed the ENCODE standards for normalized strand coefficient and relative strand correlation[72]. Peaks were then called using MACS2[73,74] with the default settings for NKX2-1, H3K4me3, and H3K27ac, and the broad setting for the broad histone mark H3K4me1. Peaks were then filtered based on the $-\log_{10} q$ value as follows: 10 for NKX2-1, 15 for H3K4me3 and H3K27ac, and 4 for H3K4me1. Subsequent peaks were then filtered for sites overlapping with the mm10 blacklist. Differentially NKX2-1 bound peaks were identified using DiffBind[75] normalized for sample read depth and at a fixed peak width of 500 bp between controls versus mutants, AT1 versus AT2, and E14.5 whole lung versus other time points in AT1 and AT2 cells. Quantification of histone marks between AT1 and AT2 cells at NKX2-1 bound sites was carried out also using DiffBind normalized for sample read depth but with a fixed peak width of 1000 bp. Foreground normalization was carried out using the fractions of reads in peaks (Frip) calculated by DiffBind for all peaks in each sample of the same antibody type. These Frip values were then multiplied by the post-filtering library read depth to scale MACS2 output bedgraph files as well as profile plots, tracks, and heatmaps in EA-seq[76]. HOMER motif analysis[77] was carried out to determine possible cofactors interacting with NKX2-1 for all peaks associated with the list. Additional motif analysis was carried out on the bottom 3000 NKX2-1 peaks accessible across all cell types with the lowest H3K4me3 average signal. NKX2-1 binding peaks were consolidated from NKX2-1 ChIP-seq at E14.5, in mature AT1 cells, and in mature AT2 cells and used for differential analysis of NKX2-1 binding sites between E14.5 and AT1 or AT2 cells. This consolidated peakset was then cross-referenced with the 10-week-old AT1 and AT2 cell differential NKX2-1 binding analysis to identify NKX2-1 peaks called only within the progenitor. These peaksets were also used to compare between E14.5 and P7 samples. Raw signal averages and $\log_2$ fold change values were computed on the top 10 or 20% for each category over time and between controls and mutants. Binding sites were annotated to genes using ChIPseeker[78].

**Omni-ATAC-seq and analysis**. The OMNI-ATAC protocol[79] was followed with minor modifications. 60,000–100,000 sorted GFP+ cells were centrifuged at 384 rcf for 5 min at 4 °C. The cell pellet was then resuspended by pipetting three times in 50 µl of cold ATAC-RSB lysis buffer (0.1% NP-40, 0.1% Tween, 0.01% Digitonin, 10 mM Tris-HCl pH 8.1, 10 mM NaCl, 3 mM MgCl$_2$) and incubated for 3 min. Then 1 ml of cold ATAC-RSB + Tween (10 mM Tris-HCl pH 8.1, 10 mM NaCl, 3 mM MgCl$_2$, 0.1% Tween) was added. The sample tube was then inverted three times and centrifuged at 384 rcf for 10 min at 4 °C. After discarding the supernatant, the pellet was resuspended in transposition mixture composed of 22.5 µl of Reaction Mix (PBS, 2.2% Digitonin, 2.2% Tween-20), 25 µl of transposase buffer (10 mM MgCl$_2$, 20 mM Tris-HCl, 20% Dimethyl Formamide), and 2.5 µl of Tn5 enzyme (NX#-TDE1, Tagment DNA Enzyme, 15027865, Illumina) and incubated at 37 °C for 30 min in a thermocycler. Samples were then purified using the MinElute PCR Purification kit (Qiagen, 29004) and eluted with 10 µl of H$_2$O. Amplification and barcoding of ATAC libraries was carried out using the Greenleaf primers for 12 cycles of PCR enrichment following the standard OMNI-ATAC amplification PCR program. A volume of 5 µl was taken after amplification to examine amplification efficiency prior to size selection. A double-sided size selection was then performed (0.5 × −1.8 × volume) using the SPRIselect reagent (Beckman Coulter, B23318). Samples were then verified for library size and absence of primer dimers by gel electrophoresis and concentration was measured using the Qubit HS dsDNA assay (Thermo Fisher Scientific, Q32851). Samples were then sequenced on the Illumina NextSeq500 or Novaseq6000 with at least 20 million 75 bp paired-end reads per sample. Reads were demultiplexed using BCL2Fastq with the setting of a --barcode-mismatches 0 or 1 and then assessed for read quality with FastQC (http://www.bioinformatics.babraham.ac.uk/projects/fastqc/), trimmed with Trimmomatic, and aligned to the UCSC mm10 reference genome by bowtie2[80,81] using default settings. After alignment, sam files were converted to bam files and filtered for unmapped reads, PCR duplicates, singletons, chimeric alignments, and low quality aligned reads using Picard MarkDuplicates and Samtools settings: –f 3 –F 4 –F 256 –F 1024 –F 2048 –q 30. Peaks were called using MACS2 settings: -q 0.05 –nomodel –shift -100 –extsize 200 --broad. Called sites for ATAC samples were filtered at a $-\log_{10} q$ value greater than 5, and sites overlapping with the mm10 blacklist were discarded. Differentially accessible sites were identified using DiffBind normalized for sample read depth and with a fixed peak width of 500 bp between control versus mutants and AT1 versus AT2. Foreground normalization was carried out using the fractions of reads in peaks (Frip) calculated by DiffBind for all peaks in each ATAC-seq sample. These Frip values were then multiplied by the post-filtering library read depth to scale MACS2 output bedgraph files and profile plots, tracks, and heatmaps in EA-seq.

**Single-cell ATAC-seq**. Cells were sorted from 7-week-old mice as described above and processed using the 3′ Library and Gel Bead Kit following the manufacturer's users guide (v2 rev D) on the Chromium Single-Cell Gene Expression Solution Platform (10X Genomics). Libraries were then sequenced using Illumina Novaseq6000 with a 50-paired-end format with index1 for 8 cycles and index2 for 16 cycles. The output was processed using Cell Ranger ATAC's functions "cellranger-atac count". Downstream analysis was performed using Seurat R package (v3)[28], Signac (v 0.1.5) (https://github.com/timoast/signac) and custom R scripts. Cells were filtered out if the unique peak counts were over 100,000 or <1000 accessible sites, lower than 25% of reads were in peaks, or they had higher than 0.025 blacklist ratio or higher than 10 nucleosome signal, or a transcription start site enrichment <2. After cell clustering, gene activities were calculated using the 2000 bp upstream and downstream from the transcription start site. The following genes activity score was used to identify cell types: *Spock2* for AT1, *Lamp3* for AT2, *Sox2* for airway (cross-referenced with *Foxj1*, *Scgb1a1*, and *Trp63*), *Plvap* for PLVAP endothelial cells, *Car4* for CAR4 endothelial cells, *Cd79a* for B cells, *Cd3e* for T cells, *Ccl4* for NK cells, *Cd9*, Ear1, and *Cd300e* for alveolar macrophages, *Cd9* and *S100a9* for neutrophils, *Cd300e* for monocytes, *Msln* for mesothelial cells, *Cdh4* and *Fgf18* for Wnt5a cells, *Acta2* and *Actc1* for alveolar smooth muscle cells, *Pdgfra* and *Meox2* for interstitial cells, and *Pdgfrb* and *Notch3* for pericytes. Clusters with most cells accessible across all genes were excluded from analysis. To classify cell types into lineages, the following genes were used in addition to cell type information: *Nkx2-1* and *Epcam* for epithelial cells, *Cdh5* for endothelial, *Runx1* for immune, and *Tbx4* for mesenchymal. After cell type and lineage identification, cell barcodes were exported to a text file for each lineage. These barcodes were used to subset the possorted_bam.bam files output from the "cellranger-atac count" function using the program Sinto into the epithelial, endothelial, immune, and mesenchymal lineages (https://github.com/timoast/sinto). Sinto was also used to randomly subset the possorted_bam.bam for each set of barcodes into two pseudo-bulk files, resulting in eight files for the four lineages. All files were then filtered for unmapped reads, PCR duplicates, singletons, chimeric alignments, and low quality aligned reads using Picard MarkDuplicates and Samtools commands: –f 3 –F 4 –F 256 –F 1024 –F 2048 –q 30. Peaks were called on full lineage sets using MACS2 commands: -q 0.05 –nomodel –shift -100 –extsize 200 --broad. Called sites for ATAC samples were filtered at a $-\log_{10} q$ value greater than 2 and sites overlapping with the mm10 blacklist were discarded. To compare accessible sites between lineages, DiffBind was used on the two filtered pseudo-bulk lineage bam files using the peaks called for each full lineage set. To verify DiffBind output, Sinto was used to subset the possorted_bam.bam into bam files for each individual cell type. They were then filtered using the same parameters above and analyzed using MACS2 with the parameters above. The Frip values generated by DiffBind for cell types and lineages were multiplied by the associated post-filtering library depth and used for foreground normalization to scale the tracks and heatmaps in EA-seq. Visualization of lineage differential accessibility analysis showed that while statistical significance was not achieved due to variations within the endothelial, immune, and mesenchymal lineages, a $\log_2$ fold change of 1 was sufficient to qualitatively distinguish epithelial lineage versus housekeeping sites.

**Statistics and reproducibility**. Cumulative binomial distribution was used to calculate significance of motif enrichment in Figs. 5a and 7b. Analysis of scRNA-seq differential gene expression in Figs. 5g and 6e and Supplementary Figs. 4c and 5b was carried out using MAST that employs a combined binomial and normal-theory likelihood ratio test to calculate statistical significance[61]. These resulting $p$ values were then adjusted using all features in the dataset in a Bonferroni correction. All confocal images are representative of at least three imaging fields of each sample and at least three sets of control and mutant lungs except for Fig. 6h and Supplementary Fig. 6c, where three control lungs and two mutant lungs were used. Hundreds to thousands of cells were quantified in each comparison.

**Reporting summary**. Further information on research design is available in the Nature Research Reporting Summary linked to this article.

## Data availability
The authors declare that all data supporting the findings of this study are available within the article and its Supplementary information files or from the corresponding author upon reasonable request. ChIP-seq, bulk ATAC-seq, scATAC-seq, and scRNA-seq data have been deposited in the NCBI Gene Expression Omnibus database under accession code GSE158205. Source data are provided with this paper.

## Code availability

Custom script for generating the figures is available as Supplementary Software File 1.

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

## Acknowledgements

We thank Drs. Elizabeth Grove and Eric Bellefroid for providing the *Wnt3a^Cre* mice, Dr. Harold Chapman for providing the *Sftpc^CreER* mice. We thank Dr. Jan Parker-Thornburg and Chad Smith at the University of Texas MD Anderson Genetically Engineered Mouse Facility for generating the *Rtkn2^CreER* mice. We thank Dr. Lisandra Villa Ellis, Dr. Margo P. Cain, Kamryn N. Gerner-Mauro, and Odemaris Narváez del Pilar in our lab for generating scRNA-seq data for the control lungs. We thank Kamryn N. Gerner-Mauro for sequencing ChIP-seq samples. The University of Texas MD Anderson Cancer Center Genetically Engineered Mouse Facility, DNA Analysis Facility, and Flow Cytometry and Cellular Imaging Core Facility are supported by the Cancer Center Support Grant (CA #16672). This work was supported by the University of Texas MD Anderson Cancer Center Retention Fund and National Institutes of Health R01HL130129 and R01HL153511 (J.C.), and Gigli City Family Endowed Scholarship, City Federation of Women's Clubs Endowed Scholarship, and National Institutes of Health F31HL139095 (D.R.L.).

## Author contributions

D.R.L. and J.C. designed research; D.R.L. performed research and analyzed data; A.M.L. generated the scATAC-seq and AT2 bulk ATAC-seq data; Y.Y. developed the Diffbind and ChIPseeker analysis pipeline; J.C. generated the *Rtkn2^CreER* mice; H.A. provided the *Sox9^CreER* mice; S.K. provided the *Nkx2-1^CKO* mice; D.R.L. and J.C. wrote the paper; all authors read and approved the paper.

## Competing interests

The authors declare no competing interests.
