## [Peer Review File · Nature Communications]

Reviewers' Comments:

Reviewer #1:

Remarks to the Author:

The paper "Lung lineage transcription factor NKX2-1 epigenetically resolves opposing cell fates in vivo" by Little et al demonstrates that the specificity of NKX2-1 to determine AT1 or AT2 fates in lung development is controlled by differential binding of co-factors, including notably YAP/TAZ. I thought that this is a really nice paper, because it illuminates how the same TF (here NKX2-1) is able to control two opposing cell-fates. The paper relies heavily on established techniques such as ChIP-Seq, scRNA-Seq and scATAC-Seq. I did not spot anything in the bioinformatic data-analysis that would be a source of concern. Overall, the paper is very well presented with excellent figures and appears to be already at a very mature stage. I only have a couple of comments:

- Fig.4a: the authors state that they compiled 12 scRNA-Seq datasets but I could not find a statement about how these datasets were merged, and indeed the justification for their merger. Are the differences displayed in Fig.4a biological or batch effects? It would perhaps help to have a Suppl.Table, listing the 12 scRNA-Seq datasets, the technology, number of cells profiled in each developmental stage. Did cells from a similar stage but profiled in different studies cluster together?

- In the last section on cell-type specific NKX2-1 mutant cells (or in Discussion), when referring to the role of NKX2-1 loss in cancer, the discussion would be more complete if they also cited the recent work by Teschendorff & Wang "Improved detection of tumor suppressor events in single-cell RNA-Seq data" NPJ Genomic Medicine 2020, where it is shown by comparing scRNA-Seq data from lung cancer cells to normal alveolar cells that NKX2-1 is inactivated in lung cancer.

Reviewer #2:

Remarks to the Author:

Little et al. present a study of developing and adult mouse lung alveolar epithelial cells in vivo. They extensively profile the binding sites of transcription factor NKX2-1 in the mouse lung, finding that Nkx2-1 epigenetically resolves opposing cell fates in vivo. Extensive epigenetic, chip-seq, and transcriptomic datasets are employed, either at bulk or single cell resolution, to characterize mouse AT2 and AT1 cells, and lineage-selective YAP/TAZ deletion studies suggest the mechanistic role of the Hippo signaling pathway as well as NKX2-1 in regulating AT2 vs AT1 programs. There is much here of interest to the lung research community, developmental biologists, and bioinformaticians alike, given the extensive datasets presented and the implications of the central findings. The major and minor concerns below are offered to make a strong story, even stronger.

Major Concerns:

1. Mouse AT1 lineage-selective labeling and Cre drivers: Deeper characterization is needed for the new genetic mouse models used, particularly the Wnt3a and Rtkn2 Cre drivers which are not common models used for AT1 lineage-specific tracing or gene deletion. For example, further immunostaining for conventional mouse AT1 protein markers (AGER, HOPX, and /or AQP5) is needed of the cells tagged with these new reporter/drivers in order to convince the skeptical reader that lineage specific AT1 labeling/deletion is indeed being achieved, as claimed. It is not enough to claim lineage specificity without showing the data, preferably with quantitation. In addition, throughout the manuscript AT1 conventional immunostaining (using markers for AT1 listed above or SFTPC or LAMP3 for AT2 cells) would markedly improve the interpretations. For example, AT2 specific immunostaining would help to convince skeptics that AT1 rather than AT2 cells are being profiled in the shown images (particularly figs 1a, 5b, 6h) rather than simply relying on the lack of AT2 marker staining to support the claim that cells are AT1s. Since Dr. Chen's team is well versed in these stains in their prior papers this should be easy enough for them to add.

Further, immunostaining or validation of RNA expression of Spock2 in AT1 cells would add to the use of that gene as the example AT1 marker being employed for each of the ChIP/ATAC datasets as it is also a relatively new AT1 marker gene.

2. Data presentation and interpretation: the well written manuscripts clearly described the flow of experiments and interpretations, making a complicated story easy to follow; however, some of the figures are difficult to interpret either due to presentation, lack of sufficient explanation, or lack of sufficient statistics, quantitation, or clarity about reproducibility. For example, the scales for each of the genes in each condition (e.g Cre+ vs Cre-) from the Chip-seq and ATAC seq data throughout appear to be different which makes interpreting the differences in peak heights between head-to-head compared conditions difficult for the reader. If there is a way to normalize these results (or use the same scale), at least within the pairs being directly compared (eg. Cre+ vs Cre-), it would make the interpretations much clearer. In addition, the module scoring used in Figure 4d, 5j, and 6i is hard to follow as currently presented. Combining the epigenetic and RNA sequencing data is incredibly interesting and valuable but this scoring is poorly explained and the graphs are thus difficult to interpret.

The interpretation of Fig 5a-e focuses on YAP/TAZ being necessary to bring NKX2-1 to AT1 specific binding sites, but the ChIP sequencing is done on bulk lungs and later supplemental data suggests that the lungs have fewer AT1 cells. Thus, it seems there is a second potential interpretation that the YAP/TAZ deletion has led to fewer AT1 cells overall through some other interaction (not directly affecting the binding of NKX2-1). The overall effect of less AT1 cells in a bulk chip-Seq analysis would also give the appearance of decreased AT1 peaks, simply based on the absence of the AT1 cells, even without altered binding. Acknowledging this second explanation or showing direct interaction between YAP/TAZ and NKX2-1 in AT1 cells would strengthen this figure and the associated text results/interpretations.

Lastly, in figure 5f, the authors interpret cell number from one single cell experiment to claim differences in the numbers of cell types. Immunostaining or FACS data from $n > 1$ would ensure reproducibility and would be far more convincing.

3. Controversies regarding a "bipotent" AT1/AT2 progenitor: the authors interpret their data from figure 4c to suggest a bipotent population exists as late as e16.5, a finding that has raised controversy in recent lung development literature. Could the authors please add discussion to the text, placing their findings in the context of the recent literature? For example, have the authors compared their interpretation of this data to others that have employed single cell RNA sequencing time series profiles to find specification of AT2 vs AT1 cells has already occurred by that stage, arguing against the existence of a bipotent progenitor at this or later stages of development? (i.e. Frank DB, Penkala IJ, Zepp JA, et al. Early lineage specification defines alveolar epithelial ontogeny in the murine lung. Proc Natl Acad Sci U S A. 2019)

4. Interpretations of altered fates and emergence of gastrointestinal programs: the observed changes in AT1 vs AT2 fates in the genetic deletion models is one of the more important mechanistic findings of the manuscript and has great importance to the field. As currently presented it is difficult to understand how the altered lineages in the mouse mutants compare to normal control lineages head to head. For example, figure 6d would be much clearer if the datasets were combined into a single UMAP plot, allowing for more direct head to head visual comparisons of mutant cell types to control normal lineages. Are the AT1-derived AT2-like cells that arise after YAP/TAZ deletion highly similar to normal AT2 cells or very different? Showing expression of the GFP lineage trace (GFP mRNA) as well as YAP/TAZ expression (e.g. by overlaying gene expression of each of these genes onto the UMAPs) would help the reader to understand what is going on in these models and would also help identify mutant cell populations. Similarly, labeling the mutant cells on the pseudo time analysis in 6f would also be helpful.

The emergence of gastrointestinal fates in the lungs of mutant mice is an important observation, if

fully presented, and follows a recently emerging literature on this topic, including work previously published by the authors; however, this finding is presented here with very minimal data shown. Figure 7 is less developed than the rest of the paper and currently seems tacked on with regard to the gastrointestinal fate claims. The few genes shown here are not convincing enough on their own. It would be better to either cut this figure or significantly further develop the gastrointestinal story without relying so much on previously published work to support the model shown in 7e. Much more data is likely available to the authors related to the aberrant gastrointestinal signature and showing that data in much more detail would help to convince readers of the interesting interpretation claimed by the authors in the present study.

Minor Concerns:

1. The authors state that "mutant AT1 cells had a larger decrease in accessibility than mutant AT2 cells at lineage sites such as *Cdh1*" but scaling on the peaks makes it difficult to see and there is no statistical analysis to support this claim. They also acknowledge that *NKX2-1* is deleted in slightly more cells in their AT1 model than the AT2 one (86% vs 82%) and given that these results are from bulk ATAC seq data this claim needs more support and statistical testing.
2. Have the authors considered statistical testing of retained markers from Fig 3f to show they are not diminishing?
3. Addition of a timepoint at e18.5 or P0 for their epigenetic analysis in Fig3 would be interesting given the clustering seen in the RNA seq dataset in figure 4 but is not necessary for publication.
4. Figure 5i appears to have a lot of genes that are overexpressed in the mutant AT2 cells but are not described, including *Scgb1a1*. Unbiased analysis of these upregulated genes rather than focusing on the few chosen AT2 genes would further their claim of these cells being "exaggerated" AT2 cells.
5. The PCA plot and subsequent model in 6j,k is difficult to see and not quite convincing.

Reviewer #3:

Remarks to the Author:

In this manuscript, Little et al. investigated the role of *NKX2-1* transcription factor in epigenetic regulation of alveolar epithelial cell fates. The authors used multiple single-cell and bulk ChIPseq and ATACseq datasets to demonstrate that cell-type-specific binding of *NKX2-1* to chromatin occurs in promoter and enhancers of alveolar epithelial type 1 (AT1)- or AT2-specific genes. Loss of *YAP* and *TAZ* redirected *NKX2-1* from its AT1-specific to AT2-specific binding sites, leading to the appearance of "transcriptionally exaggerated" AT2-like cells. *NKX2-1*-deficient AT1 and AT2 cells gained additional chromatin accessible sites, including those specific to the opposite fate while stimulating expression of GI genes. The manuscript is clearly written and easy to follow. The amount of bioinformatic data is very impressive, and the bioinformatic and statistical analyses are well performed. However, the manuscript is somewhat one-dimensional and lacks important biological insights. Specifically, there is a lack of functional, structural and biochemical data to validate epigenetic changes in AT1 and AT2 cells after deletion of *NKX2-1* or *YAP/TAZ*. While the authors use multiple genetic mouse lines, specific phenotypes in these mutant mice were not provided. It is unclear now the differential changes in accessibility of AT1/AT2 promoters/enhancers to *NKX2-1* influence the function, progenitor properties and ultrastructure of these cells. There is a disconnect between epigenetic/ gene expression changes and functional changes in alveolar epithelial cells as they undergo lung development in utero and after birth. Specific comments are provided below:

Major comments:

1. While *Rtkn2-CreER*; *Nkx2-1*^{-/-} and *Sftpc-CreER*; *Nkx2-1*^{-/-} adult mice were used to examine differential binding of NKX2-1 to chromatin in AT1 vs AT2 cells, it is unclear whether these chromatin changes translate to any functional and structural changes in the adult lung. Does NKX2-1 deletion ultimately result in changes of blood oxygenation and lung mechanics (tissue resistance, compliance, lung volumes)?
2. Does AT2-specific deletion of NKX2-1 change basic biological functions of AT2 cells, such as surfactant secretion and the ability of AT2 cells to form lung organoids on Matrigel?
3. Does AT2-specific deletion of NKX2-1 alter progenitor properties of AT2 cells during lung injury/repair in vivo?
4. The authors propose that YAP/TAZ/TEAD direct NKX2-1 to its AT1-specific sites and prevent its binding to AT2-specific sites. While these results are very interesting, this hypothesis lacks the molecular mechanism. Does any member of YAP/TAZ/TEAD pathway physically bind to NKX2-1 to facilitate the recruitment of NKX2-1 protein to DNA? Immunoprecipitation experiments can address this issue.
5. The YAP/TAZ mutants had fewer AT1 cells at embryonic day E18.5. What is the phenotype of these mice after birth? Do they have respiratory distress or defects in postnatal alveologenesis as a consequence of AT1-deficiency in utero?
6. The authors report the existence of "transcriptionally exaggerated" AT2 cells in YAP/TAZ mutants. While these "exaggerated" AT2 cells may be different on epigenetic level, there is no evidence that these epigenetic changes result in abnormalities in structure or function of AT2 cells. What are ultrastructural and functional differences between normal and mutant AT2 cells?
7. In the last part of the manuscript, the authors identified increased expression of gastrointestinal genes in mutant AT1 and AT2 cells. While these data confirm previous observations from the same and other groups, the authors did not offer any novel molecular mechanisms by which NKX2-1 regulates GI fate. The value of these results is unclear.

We thank the reviewers for their positive remarks and suggestions to further improve our manuscript, and have addressed them accordingly. The updates are marked with track-change in Word.

REVIEWER #1

I thought that this is a really nice paper... I did not spot anything in the bioinformatic data-analysis that would be a source of concern. Overall, the paper is very well presented with excellent figures and appears to be already at a very mature stage.

We thank the reviewer for evaluating our bioinformatics analysis and positive remarks.

- Fig.4a: the authors state that they compiled 12 scRNA-Seq datasets but I could not find a statement about how these datasets were merged, and indeed the justification for their merger. Are the differences displayed in Fig.4a biological or batch effects? It would perhaps help to have a Suppl.Table, listing the 12 scRNA-Seq datasets, the technology, number of cells profiled in each developmental stage. Did cells from a similar stage but profiled in different studies cluster together?

The methods now read “Each sample group, such as control versus mutant or 12-sample controls, was merged using Cell Ranger’s “cellranger count” and “cellranger aggr”, which struck a balance between ameliorating batch effects and preserving biological differences”. Fig. 4a legends now read “Cells from similar developmental stages (P4, P6, P7, and P8) are clustered together, suggesting the differences among shifted clusters are largely biological.” All scRNA-seq and scATAC-seq data were generated using 10x Genomics, as stated in Methods; the number of cells profiled in each developmental stage is now included in Fig. 4a.

- In the last section on cell-type specific NKX2-1 mutant cells (or in Discussion), when referring to the role of NKX2-1 loss in cancer, the discussion would be more complete if they also cited the recent work by Teschendorff & Wang “Improved detection of tumor suppressor events in single-cell RNA-Seq data” NPJ Genomic Medicine 2020, where it is shown by comparing scRNA-Seq data from lung cancer cells to normal alveolar cells that NKX2-1 is inactivated in lung cancer.

This reference has been added to Discussion “The conversion among AT1, AT2, and gastrointestinal fates in this study and the literature highlights remarkable cellular plasticity, even for a terminally differentiated cell type such as the AT1 cell^{9-11,24,34-36,41-43}”.

REVIEWER #2

There is much here of interest to the lung research community, developmental biologists, and bioinformaticians alike, given the extensive datasets presented and the implications of the central findings. The major and minor concerns below are offered to make a strong story, even stronger.

We thank the reviewer for the encouraging comments and thoughtful suggestions to improve the manuscript.

Major concerns:

1. Mouse AT1 lineage-selective labeling and Cre drivers: Deeper characterization is needed for the new genetic mouse models used, particularly the Wnt3a and Rtn2 Cre drivers which are not common models used for AT1 lineage-specific tracing or gene deletion. For example, further immunostaining for conventional mouse AT1 protein markers (AGER, HOPX, and /or AQP5) is needed of the cells tagged with these new reporter/drivers in order to convince the skeptical reader that lineage specific AT1 labeling/deletion is indeed being achieved, as claimed. It is not enough to claim lineage specificity without showing the data, preferably with quantitation.

Further characterization of the Wnt3a-Cre driver is now included in Supplementary Fig. 1b, showing colocalization with HOPX and AQP5, as well as the extended morphology characteristic of AT1 cells. Further characterization of the Rtn2-CreER driver is now included in Supplementary Fig. 3b, using both Rosa-Sun1GFP and Rosa-tdT reporters to show the extended morphology characteristic of AT1 cells. For quantification, the nuclear marker NKX2-1, together with the absence of LAMP3 and ECAD-outlined cell shape, is more reliable and thus used in the initial version of the manuscript.

In addition, throughout the manuscript AT1 conventional immunostaining (using markers for AT1 listed above or SFTPC or LAMP3 for AT2 cells) would markedly improve the interpretations. For example, AT2 specific immunostaining would help to convince skeptics that AT1 rather than AT2 cells are being profiled in the shown images (particularly figs 1a, 5b, 6h) rather than simply relying on the lack of AT2 marker staining to support the claim that cells are AT1s. Since Dr. Chen's team is well versed in these stains in their prior papers this should be easy enough for them to add.

In continuation of the explanation above, Fig. 1a (Wnt3a-Cre driver) is further supported by Supplementary Fig. 1b; Fig. 6h (Wnt3a-Cre Yap/Taz mutant) is further supported by Supplementary Fig. 6c, showing loss of HOPX in the mutant. For Fig. 5b (YAP/TAZ localization to AT1 cells), YAP/TAZ cannot be costained with HOPX because both antibodies are rabbit, and membrane markers AQP5 and RAGE are not useful for colocalization with the nuclear YAP/TAZ. It thus is shown with the Shh-Cre/LAMP3/ECAD combination, and substantiated with the observation that “the canonical target genes, *Ctgf* and *Cyr61*²⁷, were specific to AT1 cells (Supplementary Fig. 5a)”.

Further, immunostaining or validation of RNA expression of Spock2 in AT1 cells would add to the use of that gene as the example AT1 marker being employed for each of the ChIP/ATAC datasets as it is also a relatively new AT1 marker gene.

As SPOCK2 is predicted to be a secreted protein, we have not identified a suitable antibody for immunostaining. Rather, we have taken advantage of our 12-sample scRNA-seq data covering all major lung cell types of both epithelial and non-epithelial lineages and show that *Spock2* is restricted to AT1 cells across developmental stages (Supplementary Fig. 3e).

2. Data presentation and interpretation: the well written manuscripts clearly described the flow of experiments and interpretations, making a complicated story easy to follow; however, some of the figures are difficult to interpret either due to presentation, lack of sufficient explanation, or lack of sufficient statistics, quantitation, or clarity about reproducibility. For example, the scales for each of the genes in each condition (e.g Cre+ vs Cre-) from the Chip-seq and ATAC seq data throughout appear to be different which makes interpreting the differences in peak heights between head-to-head compared conditions difficult for the reader. If there is a way to normalize these results (or use the same scale), at least within the pairs being directly compared (eg. Cre+ vs Cre-), it would make the interpretations much clearer.

The y-axes in ChIP-seq, ATAC-seq, and scATAC-seq are scaled by read depth and Frip (Fractions of reads in peaks), and are thus supposed to be different across samples/conditions, but are directly comparable as shown in the figures. The axes are the same for different genes of the same sample/condition. The Methods read “Foreground normalization was carried out using the fractions of reads in peaks (Frip) calculated by DiffBind for all peaks in each sample of the same antibody type. These Frip values were then multiplied by the post-filtering library read depth to scale MACS2 output bedgraph files as well as profile plots, tracks, and heatmaps in EA-seq”.

Regarding quantification and statistics, they are reported in the supplementary tables. We have additionally highlighted the marked peaks in those tables.

We have clarified these points in the Fig. 1d legend “The Y-axes are scaled via foreground normalization so that peak heights can be directly compared across samples; quantification and statistics are reported in the corresponding Supplementary tables with marked peaks highlighted (same in subsequent figures)”.

In addition, the module scoring used in Figure 4d, 5i, and 6i is hard to follow as currently presented. Combining the epigenetic and RNA sequencing data is incredibly interesting and valuable but this scoring is poorly explained and the graphs are thus difficult to interpret.

The main text now reads “To buffer the known uncertainty in attributing regulatory elements to target genes²⁵, we assigned all acquired or retained cell-type-specific sites to their nearest gene and generated an expression score averaging over all genes within a set (>1,000 genes) – an approach similarly deployed in cell-cycle scoring in Seurat or gene set enrichment analysis^{26,27}”.

The interpretation of Fig 5a-e focuses on YAP/TAZ being necessary to bring NKX2-1 to AT1 specific binding sites, but the ChIP sequencing is done on bulk lungs and later supplemental data suggests that the lungs have fewer AT1 cells. Thus, it seems there is a second potential interpretation that the YAP/TAZ deletion has led to fewer AT1 cells overall through some other interaction (not directly affecting the binding of NKX2-1). The overall effect of less AT1 cells in a bulk chip-Seq analysis would also give the appearance of decreased AT1 peaks, simply based on the absence of the AT1 cells, even without altered binding. Acknowledging this second explanation or showing direct interaction between YAP/TAZ and NKX2-1 in AT1 cells would strengthen this figure and the associated text results/interpretations.

We thank the reviewer for the insightful comments. We have provided additional supporting evidence for a possible direct interaction between YAP/TAZ/TEAD and NKX2-1, and acknowledged the second interpretation.

The direct, biochemical interaction between YAP/TAZ and NKX2-1 in vivo is technically difficult. We have described additional supporting evidence: “Furthermore, 73% of the sites with decreased NKX2-1 binding due to loss of YAP/TAZ had TEAD motifs and the average distance between TEAD and NKX motifs were 52 base pairs, as opposed to 35% and 79 base pairs for unaffected sites and 23% and 97 base pairs for sites with increased NKX2-1 binding (Supplementary Table 4). These biases in the co-occurrence and spacing between TEAD and NKX motifs were consistent with the possibility that YAP/TAZ/TEAD and NKX2-1 exist in a transcription regulatory complex, as suggested by cell culture and human genetics studies^{31,32}.”

The conclusion now reads “Collectively, YAP/TAZ and by extension TEADs direct NKX2-1 to its AT1-specific sites and prevent its binding to AT2-specific sites, at least on the population level. However, we cannot exclude the possibility that other factors directly or indirectly mediate the regulation of NKX2-1 by YAP/TAZ/TEAD”.

Lastly, in figure 5f, the authors interpret cell number from one single cell experiment to claim differences in the numbers of cell types. Immunostaining or FACS data from n>1 would ensure reproducibility and would be far more convincing.

Immunostaining of independent mutants is added to show a decrease in AT1 cells and an increase in AT2 cells in Supplementary Fig. 5e.

3. Controversies regarding a “bipotent” AT1/AT2 progenitor: the authors interpret their data from figure 4c to suggest a bipotent population exists as late as e16.5, a finding that has raised controversy in recent lung development literature. Could the authors please add discussion to the text, placing their findings in the context of the recent literature? For example, have the authors compared their interpretation of this data to others that have employed single cell RNA sequencing time series profiles to find specification of AT2 vs AT1 cells has already occurred by that stage, arguing against the existence of a bipotent progenitor at this or later stages of development? (i.e. Frank DB, Penkala IJ, Zepp JA, et al. Early lineage specification defines alveolar epithelial ontogeny in the murine lung. Proc Natl Acad Sci U S A. 2019)

We have included the Frank et al. 2019 study, as well as our recent review, in the main text “Although SOX9 progenitors continuously exit branch tips and become AT1 and AT2 cell precursors at E16.5^{25,26}, this initial shift in gene signature was placed by Monocle within the progenitor branch and prior to the AT1/AT2 divergence (Fig. 4b, c).”

4. Interpretations of altered fates and emergence of gastrointestinal programs: the observed changes in AT1 vs AT2 fates in the genetic deletion models is one of the more important mechanistic findings of the manuscript and has great importance to the field. As currently presented it is difficult to understand how the altered lineages in the mouse mutants compare to normal control lineages head to head. For example, figure 6d would be much clearer if the datasets were combined into a single UMAP plot, allowing for more direct head to head visual comparisons of mutant cell types to control normal lineages. Are the AT1-derived AT2-like cells that arise after YAP/TAZ deletion highly similar to normal AT2 cells or very different? Showing expression of the GFP lineage trace (GFP mRNA) as well as YAP/TAZ expression (e.g. by overlaying gene expression of each of these genes onto the UMAPs) would help the reader to understand what is going on in these models and would also help identify mutant cell populations. Similarly, labeling the mutant cells on the pseudo time analysis in 6f would also be helpful.

“Control and mutant lungs are processed in parallel experimentally and computationally, and thus spatially comparable in the UMAP plot” (Fig. 5f and 6h legends). We opted to show each genotype on its own to avoid confusion from overlapping dots in the UMAP plots. Additional UMAP plots are provided in Supplementary Fig. 5c and 6a.

The identity of the AT2-like cells upon YAP/TAZ deletion “was substantiated by a module score analysis of genes associated with the exaggerated AT2 cells, showing a high score specifically in AT2 cells among all other major lung cell types (Supplementary Fig. 5d).”

We agree that tracking GFP and *Yap/Taz* mRNA would be extremely useful. However, unlike the *Nkx2-1* mutant we published previously, the floxed exons of *Yap/Taz* are too far from the sequenced 3' end and their deletion does not change mRNA stability. Unfortunately the *Yap/Taz* mutants used in scRNA-seq do not have a GFP reporter.

The emergence of gastrointestinal fates in the lungs of mutant mice is an important observation, if fully presented, and follows a recently emerging literature on this topic, including work previously published by the authors; however, this finding is presented here with very minimal data shown. Figure 7 is less developed than the rest of the paper and currently seems tacked on with regard to the gastrointestinal fate claims. The few genes shown here are not convincing enough on their own. It would be better to either cut this figure or significantly further develop the gastrointestinal story without relying so much on previously published work to support the model shown in 7e. Much more data is likely available to the authors related to the aberrant gastrointestinal signature and showing that data in much more detail would help to convince readers of the interesting interpretation claimed by the authors in the present study.

While we agree the findings in Figure 7 have not been fully explored, we feel increased accessibility upon *Nkx2-1* deletion is an integral part of our genomic profiling and it is useful to at least document the changes and offer our best interpretation.

We have now examined both AT1-specific and AT2-specific *Nkx2-1* mutant a later time point and showed “21 days after *Nkx2-1* deletion, NKX2-1^{R^{tkn2}} and NKX2-1^{S^{ftpc}} mutant cells shared the ability to form aberrant epithelial cell clusters and express previously validated gastrointestinal markers PIGR and TFF2¹¹ (Fig. 7d)”.

Minor concerns:

1. The authors state that “mutant AT1 cells had a larger decrease in accessibility than mutant AT2 cells at lineage sites such as *Cdh1*” but scaling on the peaks makes it difficult to see and there is no statistical analysis to support this claim. They also acknowledge that NKX2-1 is deleted in slightly more cells in their AT1 model than the AT2 one (86% vs 82%) and given that these results are from bulk ATAC seq data this claim needs more support and statistical testing.

As described earlier in this response, the y-axes are scaled via foreground normalization and the peaks are thus directly comparable. Regarding quantification and statistics, they are reported in the corresponding supplementary tables. We have additionally highlighted the marked peaks in those tables.

2. Have the authors considered statistical testing of retained markers from Fig 3f to show they are not diminishing?

As there is no Fig. 3f, we assume the reviewer refers to the fold-changes in Fig. 3a and 3b. These fold-changes are the averages of a few thousands sites with each site having a different statistics (reported in supplementary tables). Nevertheless, the averages change by less than 2-fold as indicated in the figure. Therefore we call these peaks “retained”.

3. Addition of a timepoint at e18.5 or P0 for their epigenetic analysis in Fig3 would be interesting given the clustering seen in the RNA seq dataset in figure 4 but is not necessary for publication.

We agree. It is technically challenging to efficiently and specifically label AT1 and AT2 cells at E18.5 and P0 for ChIP-seq.

4. Figure 5i appears to have a lot of genes that are overexpressed in the mutant AT2 cells but are not described, including *Scgb1a1*. Unbiased analysis of these upregulated genes rather than focusing on the few chosen AT2 genes would further their claim of these cells being “exaggerated” AT2 cells.

The identity of these exaggerated AT2 cells “was substantiated by a module score analysis of genes associated with the exaggerated AT2 cells, showing a high score specifically in AT2 cells

among all other major lung cell types (Supplementary Fig. 5d).” This is further supported by the PCA plot (Fig. 6j), as further explained below.

5. The PCA plot and subsequent model in 6j,k is difficult to see and not quite convincing.

The text is now expanded as “Notably, unsupervised principal component analysis (PCA) of all our NKX2-1 binding datasets showed that the first component (PC-1; 58% of the variance) captured the temporal changes and the second component (PC-2; 17% of the variance) captured the differences among progenitor, AT1, and AT2 cells, recapitulating the gradual differentiation of E14.5 progenitors toward the opposing AT1 and AT2 cell fates (Fig. 6j). The E18.5 Y/T^{Sox9} mutant drifted horizontally past AT2 cells, consistent with exaggerated AT2 cells; the P15 Y/T^{Wnt3a} mutant AT1 cells drifted toward AT2 cells, consistent with AT1-to-AT2 conversion (Fig. 6j).”

REVIEWER #3

The manuscript is clearly written and easy to follow. The amount of bioinformatic data is very impressive, and the bioinformatic and statistical analyses are well performed. However, the manuscript is somewhat one-dimensional and lacks important biological insights.

We thank the reviewer for pointing out the strengths and the suggestions to provide additional phenotypic analysis to justify the biological importance of our epigenetic analysis. We feel the known importance of NKX2-1 in the lung justifies further mechanistic study, which this manuscript pursues on an epigenetic level. In particular, our prior publication (Little et al. PNAS 2019) establishes a hitherto unknown role of NKX2-1 in AT1 cells, prompting our current genome-wide comparison of AT1 versus AT2-cell-specific functions of NKX2-1. We have provided detailed molecular analysis of the mutant phenotypes, and in this revision described additional morphological phenotypes.

1. While Rtnk2-CreER; Nkx2-1/- and Sftpc-CreER; Nkx2-1/- adult mice were used to examine differential binding of NKX2-1 to chromatin in AT1 vs AT2 cells, it is unclear whether these chromatin changes translate to any functional and structural changes in the adult lung. Does NKX2-1 deletion ultimately result in changes of blood oxygenation and lung mechanics (tissue resistance, compliance, lung volumes)?

We have now examined both AT1-specific and AT2-specific *Nkx2-1* mutant a later time point and showed “21 days after *Nkx2-1* deletion, NKX2-1^{Rtnk2} and NKX2-1^{Sftpc} mutant cells shared the ability to form aberrant epithelial cell clusters and express previously validated gastrointestinal markers PIGR and TFF2¹¹ (Fig. 7d)”. We feel that changes in blood oxygenation and lung mechanics are further removed from the primary, direct function of NKX2-1.

2. Does AT2-specific deletion of NKX2-1 change basic biological functions of AT2 cells, such as surfactant secretion and the ability of AT2 cells to form lung organoids on Matrigel?

Deletion of *Nkx2-1* in AT2 cells leads to loss of AT2 proteins including surfactant protein C (Fig. 2b). Thus, we expect surfactant secretion would be secondarily affected.

Deletion of *Nkx2-1* in AT2 cells leads to aberrant expression of gastrointestinal proteins (Fig. 7d). Thus, we expect organoids from mutant AT2 cells would not be lung-like.

3. Does AT2-specific deletion of NKX2-1 alter progenitor properties of AT2 cells during lung injury/repair in vivo?

AT2 cells are primarily known for surfactant production, which is examined in this manuscript, and more recently studied as progenitor/stem cells, a function manifested during injury/repair. Therefore, interpretation of an injury/repair phenotype would be confounded by the aberrant baseline function in surfactant production. In addition, we show that NKX2-1 is required for expression of both AT1 and AT2 genes. In its absence, AT2 cells should not be able to initiate expression of AT1 genes during an AT2 to AT1 conversion.

4. The authors propose that YAP/TAZ/TEAD direct NKX2-1 to its AT1-specific sites and prevent its binding to AT2-specific sites. While these results are very interesting, this hypothesis lacks the molecular mechanism. Does any member of YAP/TAZ/TEAD pathway physically bind to NKX2-1 to facilitate the recruitment of NKX2-1 protein to DNA? Immunoprecipitation experiments can address this issue.

Physical interaction between TAZ and NKX2-1 was reported in cultured cell lines (references below), but is challenging to show in vivo and it is unclear if cultured AT1/AT2 cells are the same as those in vivo. Instead we now show that NKX2-1 and TEAD motifs are only ~50 bp apart, consistent with possible physical interaction. The main text now reads “73% of the sites with decreased NKX2-1 binding due to loss of YAP/TAZ had TEAD motifs and the average distance between TEAD and NKX motifs were 52 base pairs, as opposed to 35% and 79 base pairs for unaffected sites and 23% and 97 base pairs for sites with increased NKX2-1 binding (Supplementary Table 4). These biases in the co-occurrence and spacing between TEAD and NKX motifs were consistent with the possibility that YAP/TAZ/TEAD and NKX2-1 exist in a transcription regulatory complex, as suggested by cell culture and human genetics studies^{33,34}.”

We also acknowledge in the conclusion that “Collectively, YAP/TAZ and by extension TEADs direct NKX2-1 to its AT1-specific sites and prevent its binding to AT2-specific sites, at least on the population level. However, we cannot exclude the possibility that other factors directly or indirectly mediate the regulation of NKX2-1 by YAP/TAZ/TEAD”.

5. The YAP/TAZ mutants had fewer AT1 cells at embryonic day E18.5. What is the phenotype of these mice after birth? Do they have respiratory distress or defects in postnatal alveologenesis as a consequence of AT1-deficiency in utero?

We have now included wide-field views of the Yap/Taz mutant lung at E18.5 (Supplementary Fig. 5e), showing wide-spread air space enlargement and loss of AT1 proteins. We expect such dramatic changes would interfere with respiration at birth and, secondarily, alveologenesis later in life. However, the tamoxifen used to induce embryonic Cre recombination often interferes with the normal birth process, making it challenging to examine these secondary phenotypes.

6. The authors report the existence of “transcriptionally exaggerated” AT2 cells in YAP/TAZ mutants. While these “exaggerated” AT2 cells may be different on epigenetic level, there is no evidence that these epigenetic changes result in abnormalities in structure or function of AT2 cells. What are ultrastructural and functional differences between normal and mutant AT2 cells?

The gross structural changes in the Yap/Taz mutant (Supplementary Fig. 5e) would be a confounding factor to interpret functional changes in mutant AT2 cells in vivo. Nevertheless, Yap/Taz mutant AT2 cells have been shown to be defective in an infection model (PMID 30985294) and a pneumonectomy model (PMID 27498861).

7. In the last part of the manuscript, the authors identified increased expression of gastrointestinal genes in mutant AT1 and AT2 cells. While these data confirm previous observations from the same and other groups, the authors did not offer any novel molecular mechanisms by which NKX2-1 regulates GI fate. The value of these results is unclear.

While we agree the findings in Figure 7 need to be further explored, we feel that like the decreased accessibility (Fig. 2), the increased accessibility upon *Nkx2-1* deletion is an integral part of our genomic profiling and it is useful to at least document the changes and offer our best interpretation.

We have now also examined both AT1-specific and AT2-specific *Nkx2-1* mutant a later time point and showed “21 days after *Nkx2-1* deletion, NKX2-1^{Rtkn2} and NKX2-1^{Sftpc} mutant cells shared the ability to form aberrant epithelial cell clusters and express previously validated gastrointestinal markers PIGR and TFF2¹¹ (Fig. 7d)”.

Reviewers' Comments:

Reviewer #1:

Remarks to the Author:

I had only requested 2 revisions. The authors have addressed one of them. The second one however has not been properly addressed. While I understand that the focus of the present paper is on elucidating the role of NKX2-1 in determining AT1/2 fates in lung development, I wanted to see a few sentences in Discussion highlighting that NKX2-1 appears to be a key tumor suppressor in lung cancer. I think that when discussing the increased epigenetic plasticity of NKX2-1 mutant cells, that a reference could be made to the fact that NKX2-1 loss is seen in lung cancer development and that NKX2-1 loss is likely to increase epigenetic plasticity as an early event in lung cancer development. I believe that this is an important and deep observation, and not one to be dismissed. So, I suggest that the authors do a proper job to cite the following papers in the right context.

- Teixeira VH et al Nat Med 2019
- Teschendorff & Wang, NPJ Genomic Med.2020

Reviewer #2:

Remarks to the Author:

All my concerns have been addressed in this very nice revision, and for those that were not addressed acceptable explanations were provided by the authors. I have no residual concerns.

Reviewer #3:

Remarks to the Author:

This is a revised manuscript which investigates the role of NKX2-1 transcription factor in epigenetic regulation of alveolar epithelial cell fates. Unfortunately, the authors have not addressed a majority of my comments, and the manuscript remains one-dimensional and lacks important biological insights. There is a lack of functional, structural and biochemical data to validate epigenetic changes in AT1 vs AT2 cells after deletion of NKX2-1 or YAP/TAZ. The authors were not responsive to my critique and did not clarify now the differential changes in accessibility of AT1/AT2 promoters/enhancers to NKX2-1 influence the function, progenitor properties and ultrastructure of epithelial cells. There is no link between epigenetic vs functional changes in AT2 and AT1 alveolar epithelial cells as they become fully differentiated. Specific comments are provided below:

Major comments:

1. Rtkn2CreER; Nkx2-1^{-/-} and SftpcCreER; Nkx2-1^{-/-} adult mice were used to examine differential binding of NKX2-1 to chromatin in AT1 vs AT2 cells. However, it is still unclear whether the chromatin changes in AT1 vs AT2 cells cause any functional and structural changes in the adult lung. There is no link between gene expression and functional changes in these mouse lines.
2. Does AT2-specific deletion of NKX2-1 change basic biological functions of AT2 cells, such as surfactant secretion and the ability of AT2 cells to form lung organoids on Matrigel? – This concern was completely ignored by the authors. Instead of providing direct experimental evidence, they choose to speculate about potential and “expectable” outcomes.
3. Does AT2-specific deletion of NKX2-1 alter progenitor properties of AT2 cells during lung injury/repair in vivo? – Again, the authors provided a speculation instead of experimental evidence.
4. The authors propose that YAP/TAZ/TEAD direct NKX2-1 to its AT1-specific sites and prevent its binding to AT2-specific sites. While these results are very interesting, this hypothesis completely lacks the molecular mechanism. Does any member of YAP/TAZ/TEAD pathway physically bind to NKX2-1 to facilitate the recruitment of NKX2-1 protein to DNA? Immunoprecipitation experiments

can address this issue. – The authors did not perform or even attempted these experiments. The cell culture conditions for primary AT2 and AT1 cells are well established.

5. The YAP/TAZ mutants had fewer AT1 cells at embryonic day E18.5. What is the phenotype of these mice after birth? Do they have respiratory distress or defects in postnatal alveologenesis as a consequence of AT1-deficiency in utero? – The authors did not provide satisfactory response to this comment citing the potential problems with tamoxifen treatment. To avoid these problems, the tamoxifen can be given before E15.5 or immediately after birth. Postnatal phenotypes can be examined to address this concern.

6. The authors identified increased expression of gastrointestinal genes in mutant AT1 and AT2 cells. While these data confirm previous observations from the same and other groups, the authors did not offer any novel molecular mechanisms. – After revision, the authors included an additional time-point to confirm gene expression changes. These data add very little to clarify the molecular mechanism through which NKX2-1 regulates GI fate.

Reviewer #1 (Remarks to the Author):

I had only requested 2 revisions. The authors have addressed one of them. The second one however has not been properly addressed. While I understand that the focus of the present paper is on elucidating the role of NKX2-1 in determining AT1/2 fates in lung development, I wanted to see a few sentences in Discussion highlighting that NKX2-1 appears to be a key tumor suppressor in lung cancer. I think that when discussing the increased epigenetic plasticity of NKX2-1 mutant cells, that a reference could be made to the fact that NKX2-1 loss is seen in lung cancer development and that NKX2-1 loss is likely to increase epigenetic plasticity as an early event in lung cancer development. I believe that this is an important and deep observation, and not one to be dismissed. So, I suggest that the authors do a proper job to cite the following papers in the right context.

- Teixeira VH et al Nat Med 2019

- Teschendorff & Wang, NPJ Genomic Med.2020

We thank the reviewer for the comments and additional references. We have emphasized this point in Discussion “These cellular potentials could be unleashed under pathological conditions. For example, NKX2-1 is considered a tumor suppressor in lung cancer – a function perhaps attributable to its role in limiting cellular potentials such that the increased epigenetic plasticity of NKX2-1 mutant tumor cells may allow adaptation and growth advantage in the tumor microenvironment^{49,54}. Furthermore, the observed proliferation upon NKX2-1 loss could provide the substrate for or synergize with additional oncogenes and tumor suppressors.”

Reviewer #2 (Remarks to the Author):

All my concerns have been addressed in this very nice revision, and for those that were not addressed acceptable explanations were provided by the authors. I have no residual concerns.

Reviewer #3 (Remarks to the Author):

This is a revised manuscript which investigates the role of NKX2-1 transcription factor in epigenetic regulation of alveolar epithelial cell fates. Unfortunately, the authors have not addressed a majority of my comments, and the manuscript remains one-dimensional and lacks important biological insights. There is a lack of functional, structural and biochemical data to validate epigenetic changes in AT1 vs AT2 cells after deletion of NKX2-1 or YAP/TAZ. The authors were not responsive to my critique and did not clarify now the differential changes in accessibility of AT1/AT2 promoters/enhancers to NKX2-1 influence the function, progenitor properties and ultrastructure of epithelial cells. There is no link between epigenetic vs functional

changes in AT2 and AT1 alveolar epithelial cells as they become fully differentiated. Specific comments are provided below:

We have provided additional data, reasoning, and discussions to allay this reviewer's concerns.

Major comments:

1. *Rtkn2CreER; Nkx2-1^{-/-} and SftpcCreER; Nkx2-1^{-/-} adult mice were used to examine differential binding of NKX2-1 to chromatin in AT1 vs AT2 cells. However, it is still unclear whether the chromatin changes in AT1 vs AT2 cells cause any functional and structural changes in the adult lung. There is no link between gene expression and functional changes in these mouse lines.*

The direct function of a transcription factor including NKX2-1 is to regulate transcription. Therefore, our study focuses on the associated epigenetic mechanisms and has shown cellular changes, such as surfactant gene expression and cell morphology, which are fundamental to lung function and structure. The suggested experiments (e.g. oxygen saturation and lung mechanics) measure the consequences of molecular and cellular changes, and are common in clinical practices where fresh biopsies are often unavailable so that one has to resort to non-invasive, indirect readouts of a gene's function. Furthermore, given the known differences in human and mouse physiology (e.g. 10 times faster breathing rate in mice), overemphasis on clinical measurements in animal models likely contributes to findings in mice being not always applicable to humans, even when the underlying biology is conserved.

2. *Does AT2-specific deletion of NKX2-1 change basic biological functions of AT2 cells, such as surfactant secretion and the ability of AT2 cells to form lung organoids on Matrigel? – This concern was completely ignored by the authors. Instead of providing direct experimental evidence, they choose to speculate about potential and “expectable” outcomes.*

We provided reasons for not pursuing the suggested experiments. To reiterate, if a cell stops transcribing surfactant genes, there is no surfactant to secret. The in vitro organoid assay measures AT2 cell proliferation and differentiation, both of which we have examined in vivo in this study.

3. *Does AT2-specific deletion of NKX2-1 alter progenitor properties of AT2 cells during lung injury/repair in vivo? – Again, the authors provided a speculation instead of experimental evidence.*

Our study focuses on the primary function of AT2 cells – surfactant production to reduce surface tension. The reviewer's emphasis on injury/repair and the organoid assay above

presumably arises from their interest in the progenitor function of AT2 cells, which would be largely irrelevant in a normal healthy mouse as there is little alveolar cell turnover. These progenitor-related experiments would also be difficult to interpret because NKX2-1 mutant AT2 cells lose AT2 cell identity and gain gastrointestinal genes and because their differentiation into AT1 cells would be compromised as they no longer have NKX2-1.

4. The authors propose that YAP/TAZ/TEAD direct NKX2-1 to its AT1-specific sites and prevent its binding to AT2-specific sites. While these results are very interesting, this hypothesis completely lacks the molecular mechanism. Does any member of YAP/TAZ/TEAD pathway physically bind to NKX2-1 to facilitate the recruitment of NKX2-1 protein to DNA? Immunoprecipitation experiments can address this issue. – The authors did not perform or even attempted these experiments. The cell culture conditions for primary AT2 and AT1 cells are well established.

We agree that direct protein interaction would add to our study, but consider it technically challenging to do properly and beyond the scope of the current study. We disagree that cultured primary AT1 and AT2 cells are a good system for this purpose because they do not recapitulate the in vivo states. For example, AT2 cells do not proliferate in vivo at baseline, but do proliferate in culture by design, and proliferation on its own could be associated with YAP/TAZ activation. AT1 cells in culture do not fully expand and lack proper cell-cell interactions that we hypothesize to activate YAP/TAZ.

We have added the following sentence to our discussion: “We note that, although NKX2-1 and YAP/TAZ/TEAD can exist in a complex in vitro^{33,34}, a limitation of this study to be addressed by future experiments is to examine possible biochemical interactions between NKX2-1 and its partner factors in purified cell types from native tissues.”

5. The YAP/TAZ mutants had fewer AT1 cells at embryonic day E18.5. What is the phenotype of these mice after birth? Do they have respiratory distress or defects in postnatal alveologenesis as a consequence of AT1-deficiency in utero? – The authors did not provide satisfactory response to this comment citing the potential problems with tamoxifen treatment. To avoid these problems, the tamoxifen can be given before E15.5 or immediately after birth. Postnatal phenotypes can be examined to address this concern.

The suggested timing (before E15.5 or immediately after birth) fails to specifically target the intended time window of AT1 cell specification. Nevertheless, we let some E15.5-injected dams give birth. “Strikingly, although tamoxifen interfered with pregnancy and pups were born one day overdue, all Y/T^{Sox9} mutant pups were cyanotic and dead except for one that was gasping – a sign of respiratory distress (Source Data 4).”

6. The authors identified increased expression of gastrointestinal genes in mutant AT1 and AT2 cells. While these data confirm previous observations from the same and other groups, the authors did not offer any novel molecular mechanisms. – After revision, the authors included an additional time-point to confirm gene expression changes. These data add very little to clarify the molecular mechanism through which NKX2-1 regulates GI fate.

We feel it is necessary to report both increased and decreased chromatin accessibilities upon Nkx2-1 loss. We have acknowledged that “Future time-course analyses are necessary to track the epigenetic changes associated with the cell-type-specific shifts in cell fate and to identify the associated regulators.”